# How You Say It Matters: Personalizing LLM Responses via Dual Time-Scale Closed-Loop Adaptation

## Abstract

Personalization in large language models (LLMs) is framed as a content problem focused on deciding what to retrieve, generate, or recommend. However, identical content can have different effects on the user's affective and cognitive states depending on how it is delivered, including its structure, tone, and relational style. We present an adaptation framework that addresses delivery personalization using a *fast loop* that corrects per-turn quality degradation within sessions, while a *slow loop* learns per-user priors across sessions. We evaluate 1,094 conversations across three models from Anthropic and OpenAI and show that our framework systematically differentiates the output along the targeted affective and cognitive dimensions and outperforms the unconditioned baseline across all measured quality outcomes, with effect sizes ($d = 0.61$–$1.26$). These results suggest that delivery can be modeled as a distinct axis of LLM personalization, adapting to both longer-term user patterns and changes within an interaction.

## 1 Introduction

Most research on large language model (LLM) personalization treats personalization primarily as a content problem: what information to retrieve from a user's history, what context to include in prompts, which objectives to optimize during fine-tuning, or how to align outputs through reinforcement learning from human feedback (RLHF). Although these approaches differ in mechanism, they largely focus on determining *what* content the model should generate (Zhang et al., 2025). This framing, however, underemphasizes another important aspect of user experience: *how* the content is delivered. Prior work shows that structure, fluency, tone, and framing influence what people encode, trust, believe, and retain (Meyer, 1975; McNamara et al., 1996; Fazio et al., 2015; Oppenheimer, 2006). As a result, two responses may be factually equivalent while producing substantially different user experiences.

Some deployed systems expose delivery controls through static user-defined settings, such as *ChatGPT*'s "Personality" (OpenAI, 2026) and *Claude*'s "Personalization" (Anthropic, 2026). These interfaces allow users to specify preferred interaction styles or tones. However, they require users to determine in advance what kind of response they need and how to request it. Moreover, although prior work recognizes that users differ from one another (Qiu et al., 2025; Woźniak et al., 2024; Zeng et al., 2025; Tan et al., 2024), most personalization research implicitly treats those preferences as stable traits. This leaves limited understanding of how the same person's preferences may shift across contexts, goals, and moments in time (Ong et al., 2017; Beckmann et al., 2020; Judd et al., 2024).

Consider two users asking: *"I'm thinking about leaving my job to start a business. What should I do?"* Alex wants a structured analysis with tradeoffs, quantitative reasoning, and a recommended sequence of actions. Maya, in contrast, is weighing questions of identity, family, and uncertainty, and wants those concerns acknowledged before receiving advice. The same factual response may therefore feel useful to Alex but emotionally unresponsive to Maya. This illustrates the *between-person* dimension of delivery personalization.

The *within-person* dimension is equally important. Suppose Alex later asks: *"What should I do about my mortgage refinance?"* On Monday morning, after reviewing financial documents, he may prefer rates, thresholds, and a direct recommendation. On Friday evening, after a difficult day he never explicitly mentions,

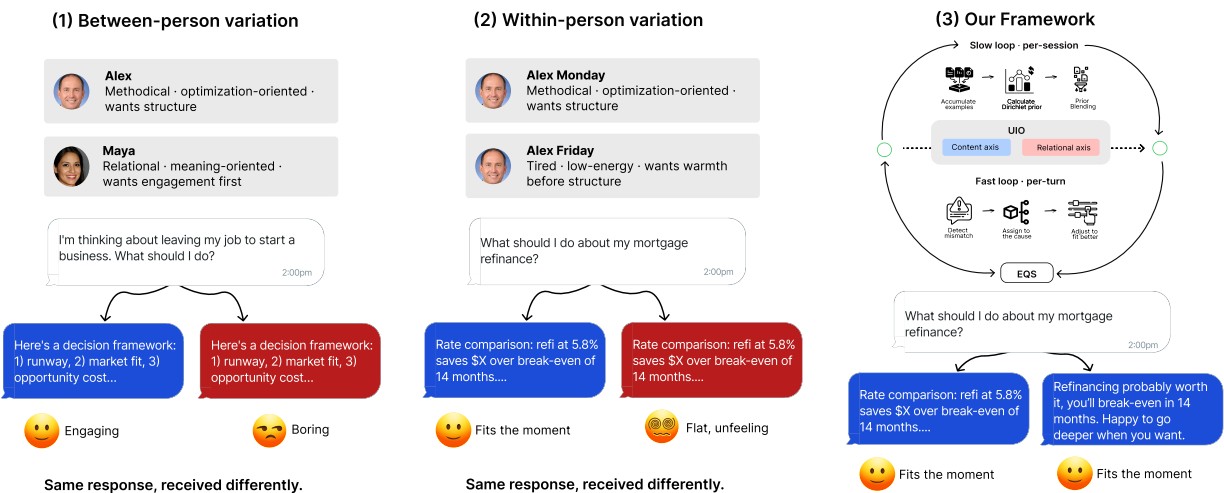

Figure 1: **Between- and within-person variation in delivery personalization. (1) Between-person variation:** Alex and Maya ask the same question, but the same response engages one and fails the other. **(2) Within-person variation:** the same user asks the same question at different moments, but a response that fits once may fail later. **(3) Our framework:** we parameterize delivery with a *UIO* over content and relational dimensions. A *slow loop* learns per-user priors across sessions, while a *fast loop* responds to per-turn failures within a session. Both are driven by a shared reward, the EQS.

the same concise response may feel poorly calibrated. The issue is not that Alex has become a different user. Rather, the response no longer matches his current state. Systems that model only stable preferences risk personalizing toward a long-term average that fails in the moment.

To address this problem, we explicitly model *delivery* as part of personalization through a unified instruction schema called the **Unified Instruction Object** (UIO). The UIO parameterizes content-related dimensions such as structure, reading level, explanation depth, and interactional form, together with relational dimensions including warmth, supportiveness, directiveness, and emotional attunement. We then adapt these dimensions through two complementary feedback loops. A **fast loop** operates within a session by detecting quality degradation, attributing failures to specific delivery dimensions, and adjusting subsequent responses. A **slow loop** operates across sessions by learning per-user Dirichlet priors over delivery dimensions and blending them with rule-based defaults at the start of future interactions. Both loops are optimized using a shared reward signal, the **Experience Quality Score** (EQS), which combines instruction compliance, behavioral engagement, and subjective feedback.

We evaluate the framework in a simulation-based study consisting of 1,094 single- and multi-turn conversations generated by pairing DEAP-grounded user states with PRISM prompts across models from Anthropic and OpenAI. We compare the full adaptive framework against both a conditioned-only ablation and an unconditioned baseline. Structured delivery conditioning makes outputs systematically distinguishable across affective and cognitive states, increasing mode-effect sizes by $1.6$–$65\times$ relative to the unconditioned baseline. The full framework improves performance across every measured quality dimension, achieving large effect sizes ($d = 0.61$–$1.26$). The fast loop targets the failing delivery dimension in 96–98% of adjustments, while the slow loop learns per-user priors that diverge monotonically from population defaults.

The key contributions of this work are as follows:

- We formulate delivery as a personalization target distinct from task content, covering both structural and relational properties of LLM responses.

- We introduce a dual time-scale inference-time adaptation framework consisting of a fast within-session correction loop and a slow cross-session preference-learning loop.

- We propose the Experience Quality Score (EQS), a unified reward signal that combines instruction compliance, behavioral engagement, and subjective feedback.

- We demonstrate across single- and multi-turn conversations that structured delivery conditioning and adaptive feedback improve interaction quality relative to conditioned-only and unconditioned baselines.

## 2    Related Work

**LLM personalization.**    LLM personalization is commonly approached through retrieval, prompting, representation learning, or feedback-based alignment  (Zhang et al., 2025). **Retrieval-based methods** store user information externally and retrieve it at inference time  (Salemi et al., 2024; Mysore et al., 2024; Li et al., 2025); **prompting methods** place user profiles or histories directly in context  (Qiu et al., 2025; Huang et al., 2024; Woźniak et al., 2024); **representation-learning** methods encode preferences into model parameters or adapters  (Liu et al., 2025; Magister et al., 2025; Tan et al., 2024); and **RLHF-based methods** use user feedback as an alignment signal  (Li et al., 2024; Jang et al., 2023; Park et al., 2024). These approaches differ in where user information is stored and how it is used, but they primarily personalize what information is selected, represented, or optimized. Instead, our work treats response delivery itself as the object of adaptation.

**User states in LLMs.**    Recent work studies whether LLMs can infer affective, cognitive, and social signals from interaction. Affective work examines emotion recognition and related enhancements  (Feng et al., 2024; Lecourt et al., 2025; Hong et al., 2025), while cognitive and social work studies personality inference and theory-of-mind reasoning  (Peters & Matz, 2024; Cursi et al., 2025; Kosinski, 2024; Street et al., 2026). These studies show that user-state signals can be estimated or modeled, but they often stop at inference. Our work asks how such inferred states should change the response format.

**Delivery and communication.**    Research in linguistics, cognitive psychology, and instructional design shows that form affects how messages are processed. Organization and positioning shape recall and learning (Meyer, 1975; Mar et al., 2021; McNamara et al., 1996), especially under working-memory constraints (Sweller, 1988; Sweller et al., 1998). Fluency also serves as a metacognitive cue for truth, credibility, coherence, and perceived intelligence  (Hasher et al., 1977; Fazio et al., 2015; Topolinski & Strack, 2009; Oppenheimer, 2006). More broadly, language conveys not only propositional content but also interpersonal stance and social context (Halliday, 1978), and speakers adapt style across situations (Giles, 1973; Giles & Ogay, 2007). This work motivates treating delivery as part of the communicative function of a response, rather than as a purely stylistic layer.

**Between- and within-person variability.**    Personalization usually assumes stable differences across users. Indeed, people differ in cognitive ability, decision-making, emotional complexity, and personality (Judd et al., 2024; Boogert et al., 2018; Pittaras et al., 2022; Ong et al., 2017; Beckmann et al., 2020). However, individuals also vary across time and context  (Siegler, 2006; Ong et al., 2017; Beckmann et al., 2020; Haegens et al., 2014; Larsen-Freeman, 2006). Because between-person and within-person variability are distinct and stable sources of variation (Judd et al., 2024; Kelava et al., 2022; Goregliad Fjaellingsdal et al., 2025; Marciano & Yeshurun, 2017), they require different design responses  (Wang et al., 2012; Whitaker et al., 2026; Pescuma et al., 2023). For LLMs, this means that a fixed user profile may capture stable preferences but miss momentary needs. This distinction motivates our separation between slow cross-session priors and fast within-session adjustment.

## 3    Methodology

We describe the framework in four components. We first define the delivery instruction space (§3.1), then describe the response generation procedure (§3.2), the reward used to score each turn (§3.3), and the two feedback loops that update delivery within and across sessions (§3.4).

### 3.1 Instruction Space and Resolution

**Delivery instruction space.** We represent response delivery with a unified instruction object (UIO): a set of structured constraints that specifies *how* a response should be expressed. The UIO has two axes. The *content axis* controls the form of explanation, including structure, reading level, technicality, depth, density, examples, and interactional shape. The *relational axis* controls the interpersonal stance of the response, including supportiveness, directiveness, warmth, emotional attunement, role stance, and related modulation. Together, these axes define 23 delivery dimensions. The full dimension set is reported in Appendix B.

**User-state representation.** The resolver does not infer user state directly. Instead, it consumes a typed `UserState` record produced by an upstream state-estimation step (Figure 2). Each state variable is stored with an inferred value, a confidence score, and its deviation from the user's baseline. This separates state inference from response control: the same resolver can consume state estimates from text-derived or other signals when available. In our experiments, the state record contains affective state, represented as valence and arousal, cognitive load, and data quality.

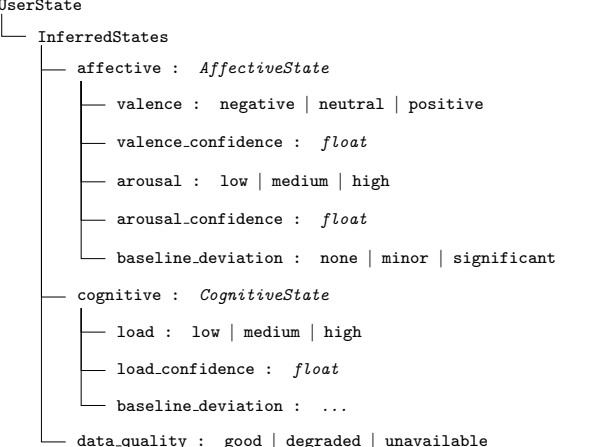

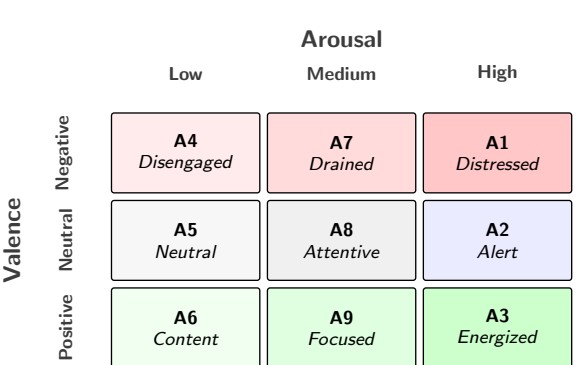

Figure 2: Schema of the `UserState` record consumed by the resolver. Each inferred state contains a value, confidence, and baseline-deviation indicator; `data_quality` records signal reliability.

Figure 3: Affective mode grid used by the resolver. Valence and arousal are mapped to modes A1–A9, which initialize the delivery policy before load, prior, and feedback adjustments.

**State-conditioned resolution.** Given a `UserState`, the resolver maps the current user state to a concrete UIO. Resolution proceeds in stages (Algorithm 1). First, the resolver checks whether the state signal is usable. If no reliable signal is available, it returns a safe default rather than personalizing from noise. It then maps valence and arousal to an affective mode using Russell's circumplex model (Russell, 1980). This mode initializes the base delivery policy: for example, negative-valence modes increase supportiveness and emotional attunement, while neutral modes use a more informational stance.

The resolver then modulates this base policy using cognitive load, blends in slow-loop user priors when enough cross-session evidence is available, and applies any fast-loop adjustment from the previous turn. The final stage clips extreme settings when confidence is low or data quality is degraded. Thus, personalization is state-conditioned but conservative: uncertain signals move the response toward neutral defaults, while stronger and repeated evidence allows more specific delivery adaptation.

### 3.2 Response Generation

We generate each response in two stages. First, we plan how the response should be delivered. Given the resolved UIO and the user prompt, the model produces an internal *ReasoningBlock*: a short, structured summary of the task, the user's likely interaction need, and the delivery constraints that matter for the

---

**Algorithm 1** UIO resolution pipeline

---

**Require:** UserState $h$, SlowLoopOutput $\ell$, optional AdjustmentSignal $a$
**Ensure:** UnifiedInstructionObject $u$

1: **function** RESOLVEUIO($h, \ell, a$)
2:     **S0: if** $h$.data_quality $=$ UNAVAILABLE **then return** SAFEDEFAULTS
3:     **S1:** Estimate affective, cognitive, and joint confidence
4:     **S2:** Map valence and arousal to affective mode $m$
5:     **S3:** $(\pi_0, p_0) \leftarrow$ RESOLVEPERSONAPOLICY($m$)
6:     **S4:** $(\pi_1, p_1) \leftarrow$ MODULATELOAD($\pi_0, p_0, h$.load)
7:     **S5:** $(\pi_2, p_2) \leftarrow$ BLENDPRIORS($\pi_1, p_1, \ell.\hat{P}_u, \ell.\alpha$)
8:     **S6: if** $a \neq \varnothing$ and $a$.should_adapt and not $a$.execution_failure **then** $(\pi_3, p_3) \leftarrow$ APPLYADJUSTMENT($\pi_2, p_2, a$)
9:         **else** $(\pi_3, p_3) \leftarrow (\pi_2, p_2)$
10:     **S7:** $u \leftarrow$ CLIPBOUNDS($\pi_3, p_3, h$.data_quality)
11:     **return** $u$
12: **end function**

---

current turn. We do not expose this block to the user. We store it only as an audit artifact, so that we can later inspect whether the model recognized the situation it was responding to.

Second, we generate the user-facing response. This call receives the user message, the resolved UIO, the ReasoningBlock, and the last $k$ conversation turns. The goal is to make delivery planning explicit before surface realization: the model first decides what kind of response would fit the user and context, then writes the response under those delivery constraints. This separation lets us control *how* the answer is expressed without changing the task content itself. Prompt templates and decoding settings appear in Appendix C.

### 3.3 Experience Quality Score

We use the **Experience Quality Score** (EQS) as the shared reward signal for both feedback loops. Our goal is to measure not only whether a response was correct, but whether its delivery fit the user and the moment in which it was produced. For each turn, we combine three sources of evidence. First, we measure **compliance**: whether the response realized the delivery instructions specified by the UIO. We evaluate compliance by intent rather than surface form, since the same delivery constraint can be satisfied in more than one wording. Second, we measure **engagement** from the user's next message, normalized against that user's own session baseline. This captures whether the user continues, elaborates, asks a more complex follow-up, or explicitly corrects the response; correction signals are capped because the absence of a correction is not itself evidence of success. Third, when available, we include **subjective feedback** from user ratings collected periodically, after detected degradation, and at session end. These ratings use eight 5-point Likert items, normalized and inverted where needed, together with two free-text retrospective questions. We do not impute missing evidence. If a component is unavailable for a turn, we drop it and renormalize the remaining weights. Thus, EQS remains defined across ordinary interaction traces while still incorporating direct user judgment whenever it is available. The full subjective item inventory and feedback schedule appear in Appendix D.

### 3.4 Dual Time-Scale Feedback

We adapt the UIO on two coupled timescales. The *fast loop* responds to failures within a session by adjusting the next turn, while the *slow loop* learns more stable delivery preferences across sessions and uses them to initialize future UIO resolution. We couple the loops in one direction: the fast loop produces within-session corrections and positive examples, and the slow loop uses those examples at session end to update the priors used in later sessions.

### 3.4.1 Quality Measurement

We measure response quality using both deterministic features and semantic evaluation. First, a deterministic analyzer built with spaCy (Honnibal et al., 2020) and `textstat` (textstat, 2026) extracts structural and linguistic features, including length, paragraph structure, readability, lexical diversity, and grammatical markers. These metrics provide stable inputs for compliance checks and fallback attribution.

Second, the SFE acts as a stateless post-hoc evaluator independent of the generator. It consumes the user message, model response, UIO, ReasoningBlock, and recent history, then returns structured judgments for safety, semantic compliance, and adaptation. The SFE can be implemented as either a trained compliance classifier or an LLM critic. We use the LLM critic because it requires no task-specific training data and generalizes across UIO dimensions. This semantic signal is used for fast-loop adaptation because it supports failure attribution, rather than only failure detection. Full prompts and settings appear in Appendix E.

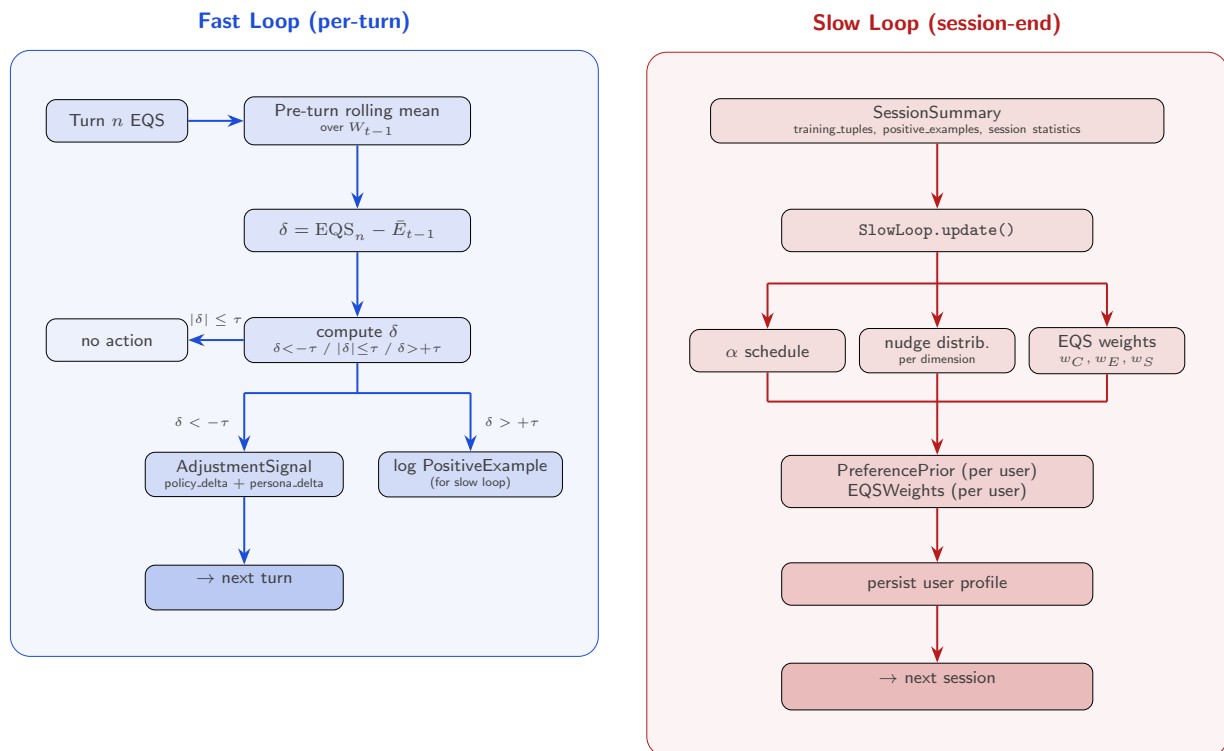

Figure 4: Dual feedback loops. The fast loop detects turn-level quality drops from EQS and SFE feedback, then adjusts the next response. The slow loop updates user priors and EQS weights at session end, subject to cross-session consent.

### 3.4.2 Fast Loop

The fast loop (Figure 4, Left) operates at turn granularity. At each turn, it compares the current EQS to a rolling baseline over previous turns. If the drop exceeds a threshold $\tau$, the turn is treated as degrading; if the increase exceeds $\tau$, the turn is treated as improving. When adaptation is needed, the loop first uses the SFE recommendation. A fixed lexicon maps SFE feedback to UIO dimensions and directions. If the semantic evaluator provides no actionable recommendation, the loop falls back to EQS attribution: low compliance indicates an execution failure, low engagement points to the weakest behavioral sub-signal, and low subjective feedback points to the worst rated dimension. Adjustments are categorical (`increase`, `decrease`, or `maintain`) and are applied as a single ordinal step. We cap each update at two target dimensions to avoid over-correction.

On the second consecutive degrading turn, the loop asks a short Likert question to disambiguate the failure before the session ends. Improving turns are logged as positive examples for the slow loop.

### 3.4.3 Slow Loop

The slow loop (Figure 4, Right) learns stable delivery preferences across sessions. It takes positive examples and session-level EQS summaries as input, then updates per-dimension preference priors and EQS component weights for the next session.

**Prior representation.** For each user, the slow loop maintains a *PreferencePrior* with one categorical distribution for each adaptable dimension. We track 14 dimensions in total: 7 policy dimensions and 7 persona dimensions (Table 1). For each dimension $d \in \mathcal{D}$ with value set $\mathcal{V}_d$, we initialize a symmetric Dirichlet prior:

$$P_u^{(d)} \sim \mathrm{Dir}\left(\alpha_0 \cdot \mathbf{1}_{|\mathcal{V}_d|}\right), \qquad \alpha_0 = 1. \tag{1}$$

This prior is non-informative, so the rule-based resolver controls cold-start behavior until evidence accumulates.

Table 1: Dimensions tracked by the slow-loop preference prior.

| Policy (7) | Persona (7) |
|---|---|
| reading_level | challenge_level |
| sentence_complexity | supportiveness |
| technicality | directiveness |
| explanation_depth | warmth |
| information_density | error_tolerance |
| examples_frequency | emotional_attunement |
| choice_complexity | interactivity |

**Example selection.** At session end, the slow loop selects training evidence in two tiers. Tier 1 contains improving turns from the fast loop whose EQS exceeds the behavioral threshold $\eta_{\mathrm{beh}}$. Tier 2 contains other high-reward turns, ranked by EQS and subjective feedback when available. Tier 2 examples enter with half weight, $\lambda_t = 0.5$. If no turn passes the threshold, we keep the single best turn with positive reward.

**Prior update.** To handle non-stationary preferences, we update priors using only a sliding window of the most recent $W$ sessions. Let $S$ be the current session index. The fractional count for value $v$ of dimension $d$ is

$$n_v^{(d,W,S)} = \sum_{s=\max(1,\,S-W+1)}^{S} \sum_{t \in \mathcal{T}_s^+} \lambda_t \cdot \mathbb{1}\left[v_t^{(d)} = v\right]. \tag{2}$$

where $\mathcal{T}_s^+$ is the set of selected turns from session $s$ and $\lambda_t \in \{1.0,\ 0.5\}$ is the tier weight. Let $N^{(d,W,S)} = \sum_v n_v^{(d,W,S)}$. Dirichlet–Categorical conjugacy gives the posterior mean:

$$\hat{P}_u(v \mid d, S) = \frac{\alpha_0 + n_v^{(d,W,S)}}{|\mathcal{V}_d| \cdot \alpha_0 + N^{(d,W,S)}}. \tag{3}$$

**Prior blending.** The influence of the learned prior grows with posterior concentration:

$$\alpha_{\mathrm{eff}}^{(d,S)} = \min\left(\frac{N^{(d,W,S)}}{|\mathcal{V}_d| \cdot \alpha_0 + N^{(d,W,S)}}, \alpha_{\max}\right). \tag{4}$$

At the start of the next session, the resolver blends the rule-based default $v_{\mathrm{rule}}$ with the learned prior:

$$v^* = \arg\max_{v \in \mathcal{V}_d}\left[\left(1 - \alpha_{\mathrm{eff}}^{(d,S)}\right) \cdot \mathbb{1}[v = v_{\mathrm{rule}}] + \alpha_{\mathrm{eff}}^{(d,S)} \cdot \hat{P}_u(v \mid d, S)\right]. \tag{5}$$

Thus, dimensions with more evidence personalize faster, while sparsely-observed dimensions remain rule-driven. We cap $\alpha_{\max} = 0.8$, preserving at least 20% weight for rule-based defaults and preventing user priors from overriding safety-gated policy bounds.

**EQS weight adaptation.** The slow loop also updates the user's EQS component weights $(w_C, w_E, w_S)$. At session end, it treats the mean subjective score as the target and increases the weight of the objective component, compliance or engagement, that best tracks it. The remaining active weights are reduced, and the vector is projected back onto the simplex with bounds $[0.1, 0.6]$. If no subjective feedback is available, the active weights are simply renormalized.

**Cold start and consent.** In the first session, the posterior is uniform and $\alpha_{\text{eff}} = 0$, so the UIO is resolved entirely by the rule-based defaults. Cross-session personalization begins in session 2 and only when the user has granted explicit consent. Without consent, no cross-session preference state is stored.

## 4 Research Questions

We evaluate whether delivery can be controlled, whether it improves measured interaction quality, and whether adaptation occurs on the two timescales introduced above.

**RQ1: Delivery controllability.** Does a structured delivery instruction make model outputs systematically different across target affective and cognitive states, relative to an unconditioned baseline?

**RQ2: Interaction quality.** Does adaptive delivery improve measured interaction quality over both a conditioned non-adaptive system and an unconditioned baseline?

**RQ3: Dual time-scale adaptation.** Do the two feedback loops act on their intended timescales: fast within-session recovery after degradation, and slow cross-session learning of user-specific priors?

## 5 Experiments

We evaluate the framework with simulated multi-state users built from two public datasets. **DEAP** (Koelstra et al., 2012) provides affective state labels, and **PRISM** (Kirk et al., 2024) provides user prompts. We discretize DEAP into a 3×3 valence–arousal grid ($A_1$–$A_9$) and select six target modes covering the poles, a neutral anchor, and three contrastive cells. We select eight subjects using variance-filtered greedy coverage and reserve one held-out distressed subject (S13, persistent $A_1$) for Run D. We filter and classify PRISM prompts into a 12-cell affect-by-task grid, yielding 116 prompts.

We compare three conditions. The full system ($C_0$) uses UIO conditioning with both feedback loops enabled. The conditioned-only ablation ($C_2$) uses the same UIO conditioning but disables both loops. The unconditioned baseline ($C_4$) bypasses the pipeline and generates directly from the user prompt.

The evaluation consists of four runs. **Runs A/B** contain 288 matched single-turn trials each and compare $C_0$ against $C_4$, isolating the effect of structured delivery conditioning. **Run D** contains 40 sessions and 259 turns, comparing the full system ($C_0$) with the conditioned-only ablation ($C_2$) in multi-session interaction. **Run D\*** replays the Run D traces through the ChatGPT and Claude chat interfaces to construct the multi-session $C_4$ arm. Subject selection, prompt filtering, condition definitions, and run protocols appear in Appendices G, H, F, and I.

## 6 Results

### 6.1 RQ1: Delivery Controllability

Structured UIO conditioning makes delivery controllable. A MANOVA over the nine-feature delivery vector shows large affective-mode effects under $C_0$ for both Claude and GPT (Claude $\eta_p^2$=0.194; GPT $\eta_p^2$=0.202; both $p$<.001). These effects are more than three times larger than under $C_4$ (Claude 0.049; GPT 0.067). The amplification holds for all nine features on both models (Fig. 5). For example, word count rises from roughly 60 tokens in $A_1$ to roughly 225 tokens in $A_9$ under $C_0$, while remaining comparatively flat under $C_4$ (Fig. 6).

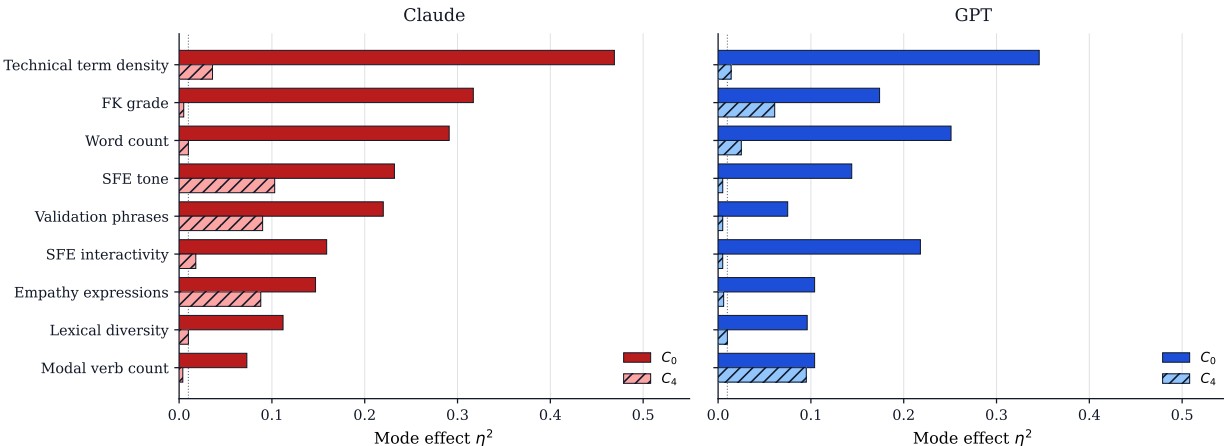

Figure 5: Affective-mode effect sizes for nine delivery features under UIO conditioning ($C_0$) and the unconditioned baseline ($C_4$). Across both models, every feature shows a larger mode effect under $C_0$. Features are ordered by Claude $C_0$ effect size. The dotted line marks $\eta^2$=0.01.

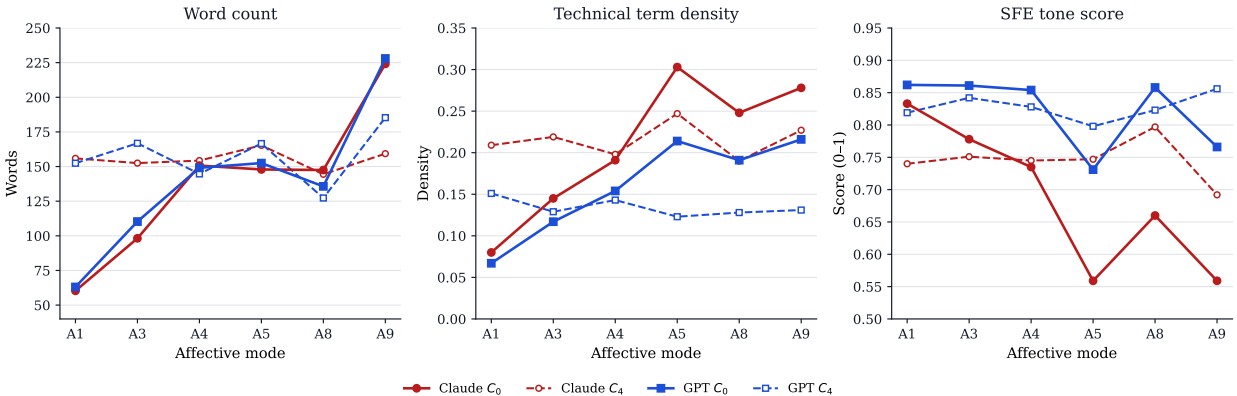

Figure 6: Mode trajectories for three discriminating delivery features. Under $C_0$, both models produce ordered shifts across target modes. Under $C_4$, the same features remain comparatively flat.

Compliance is strongest for directly specified structural constraints: paragraph count, sentence count, and reading level all exceed 92% on both models. More semantic dimensions are harder to control. Technicality reaches 50.0% compliance on Claude and 56.0% on GPT, while SFE overall fit reaches 0.735 and 0.839, respectively. The same difficulty ordering appears on both models (encouragement < tone < interactivity < challenge level). Cognitive-load conditioning follows the same pattern, with substantially larger effects under $C_0$ than $C_4$ (Claude/GPT: 0.345/0.312 vs. 0.069/0.077). The effect is also model-robust: per-feature $C_0$ effect sizes correlate strongly across Claude and GPT ($r$=0.80), and every feature satisfies $C_0$>$C_4$. Full statistics appear in Appendix J.

## 6.2 RQ2: Adaptive Delivery and Interaction Quality

Adaptive delivery improves interaction quality beyond both baselines. Pooled across models ($N$=80 paired sessions), the full system ($C_0$) outperforms the unconditioned baseline ($C_4$) on all eight outcome features. Effects are large for EQS ($d$=1.21), SFE overall fit ($d$=1.26), and all six SFE dimensions ($d$=0.61–1.15; all Holm-corrected $p$<.001; Fig. 7).

The conditioned-only ablation ($C_2$) also outperforms $C_4$ on every feature, showing that state-conditioned delivery accounts for most of the gain. The full system still improves over $C_2$ on EQS, overall fit, interactivity,

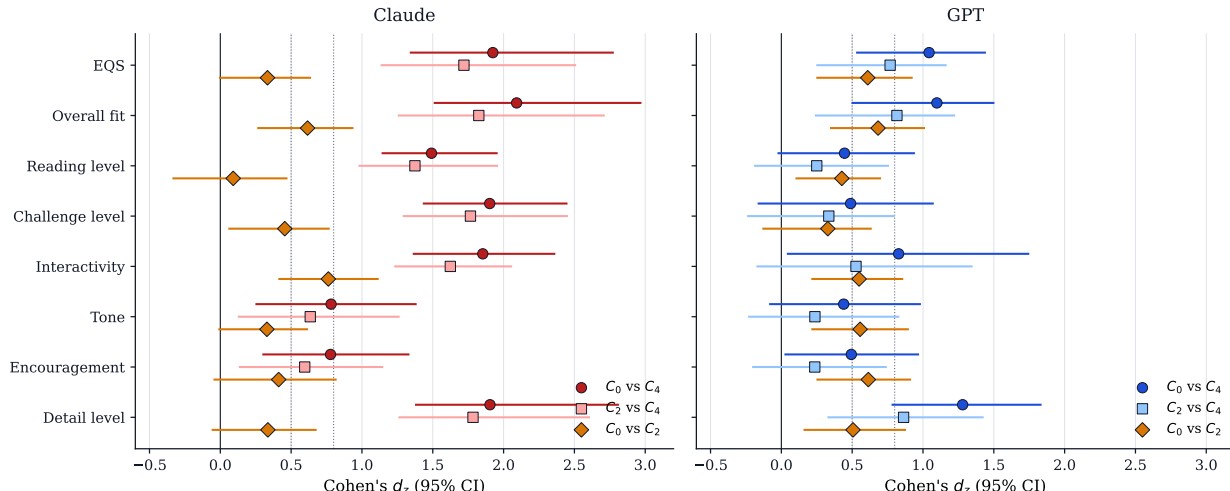

Figure 7: Pairwise condition effects across the eight RQ2 outcome features. Points show paired Cohen's $d_z$ with BCa stratified-bootstrap 95% CIs. The $C_2 - C_4$ contrast estimates the contribution of state conditioning; $C_0 - C_2$ estimates the added contribution of closed-loop adaptation.

tone, encouragement, and detail level, showing that closed-loop adaptation adds a consistent residual benefit. Decomposition analysis attributes 74–91% of the total $C_0 - C_4$ gap to conditioning and the remaining 9–26% to feedback loops. Loop gains are largest on relational dimensions, especially encouragement (26%) and tone (21%). The $C_0 > C_4$ result is robust to all pre-registered sensitivity checks, including excluding the distressed subject $S_{13}$, model-specific analyses, Bonferroni correction, and removing subjective-feedback sessions. Full comparisons and sensitivity analyses appear in Appendix K.

### 6.3 RQ3: Dual Time-Scale Adaptation

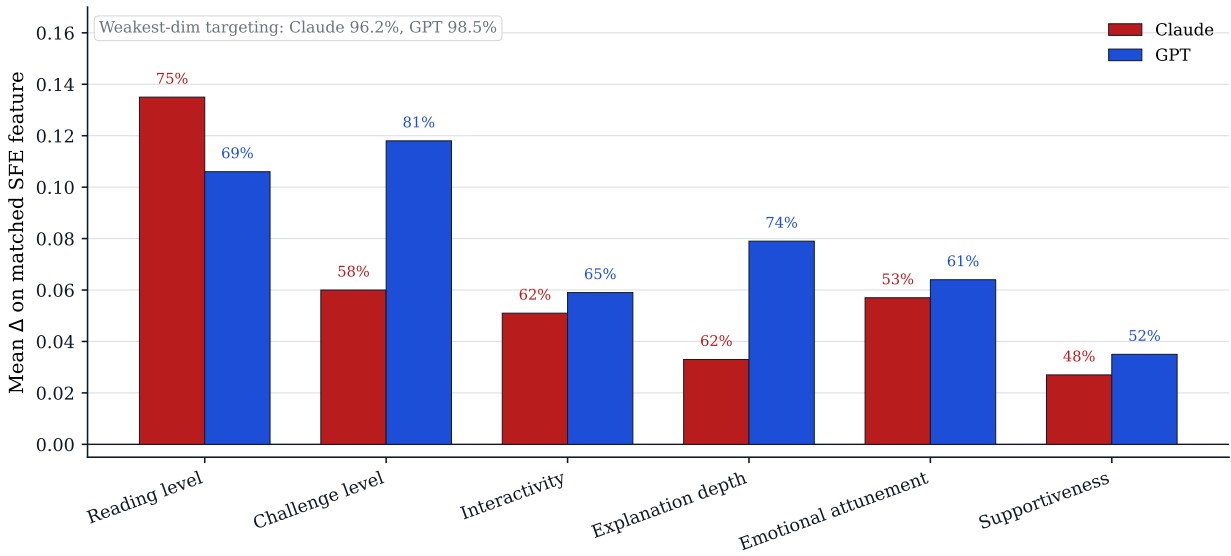

Figure 8: Fast-loop recovery by targeted dimension. Bars show mean next-turn change on the matched SFE feature after an adjustment. Labels show the percentage of adjustments with positive recovery. The inset reports how often the loop targeted the lowest-scoring SFE dimension.

The fast loop reliably localizes within-session degradation, targeting the lowest-scoring SFE dimension in 96.2% of Claude adjustments and 98.5% of GPT adjustments. Next-turn recovery is positive for nearly all targeted dimensions. Gains are largest for directly specified structural dimensions, such as `reading_level` and `challenge_level` (mean $\Delta = +0.06$ to $+0.14$), and smaller but consistent for relational dimensions, such as `supportiveness` and `emotional_attunement` (mean $\Delta = +0.03$ to $+0.06$; Fig. 8).

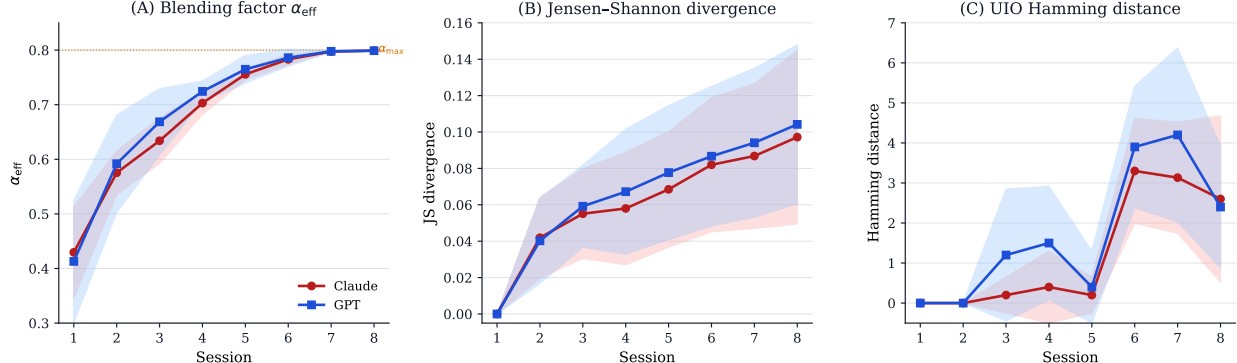

Figure 9: Slow-loop personalization across eight sessions. **(A)** The blending factor $\alpha_{\text{eff}}$ approaches the cap $\alpha_{\text{max}}$=0.8. **(B)** JS divergence from the uniform prior increases. **(C)** Hamming distance between the conditioning-only and prior-blended UIOs increases on slow-only turns. Shaded bands show $\pm 1$ SD across subjects.

The slow loop produces monotonic cross-session personalization. Across eight sessions, $\alpha_{\text{eff}}$ rises to the design cap of 0.80, JS divergence from the uniform prior grows to 0.097 on Claude and 0.104 on GPT, and the Hamming distance between the conditioning-only and prior-blended UIOs increases significantly on both models (all linear trends $p<.001$; Fig. 9). Terminal priors also diverge across users: mean between-subject Hamming distance reaches 5.0 out of 14 dimensions on Claude and 4.4 on GPT.

Personalization is concentrated rather than global. Users diverge most on relational and depth-related dimensions, including `challenge_level`, `explanation_depth`, and `emotional_attunement`. They converge more often on `technicality`, `examples_frequency`, and `choice_complexity`. Full targeting, recovery, trajectory, and divergence analyses appear in Appendix L.

# 7 Conclusion

We presented a dual time-scale framework for personalizing LLM response delivery. The results support treating delivery as a distinct target for personalization, with fast adaptation for within-session shifts and slow adaptation for cross-session preferences.

# 8 Limitations

**First**, we evaluate adaptation with simulated users rather than recruited participants. This gives us controlled state changes and matched comparisons, but it cannot fully capture how real users perceive or respond to delivery adaptation. The engagement and subjective feedback signals therefore reflect the simulator's assumptions. Our results should be read as evidence that the framework can control and adapt delivery under a grounded simulation, not as a final claim about real user experience.

**Second**, the framework relies on LLM calls for private reasoning and semantic feedback. This improves flexibility across tasks and delivery dimensions, but it adds cost and latency. The SFE also inherits the limitations of LLM-based evaluation, including possible model dependence and imperfect calibration. We mitigate this by separating generation from evaluation and by using deterministic metrics where possible, but future work should compare LLM critics with trained or hybrid evaluators.

**Third**, we focus on delivery quality rather than task correctness. The framework is intended to adapt how content is expressed, but we do not fully test whether these adaptations affect factual accuracy, reasoning quality, or downstream task success. This matters especially in high-stakes settings.

**Finally**, our evaluation covers a limited set of models, prompts, and state variables. We instantiate state with affective mode and cognitive load, but other settings may require additional signals and stricter safety constraints. Real deployments would also require consent, privacy safeguards, user controls, and mechanisms for inspecting or resetting learned delivery priors.

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

# Additional Details

Table 2: Hyperparameters.

| Component | Symbol | Value | Role |
|---|---|---|---|
| Fast loop | $\tau$ | 0.1 | EQS deviation threshold for marking a turn as degrading or improving. |
| | $|\mathcal{W}|$ | 5 | Number of previous turns used to compute the rolling EQS baseline. |
| | $m_{\max}$ | 2 | Maximum number of UIO dimensions adjusted after a degrading turn. |
| Generation | $k$ | 10 | Number of previous conversation turns supplied to the generation call. |
| EQS | $(w_C, w_E, w_S)$ | $(0.33, 0.33, 0.34)$ | Default weights for compliance, engagement, and subjective feedback. Missing components are dropped and the remaining weights are renormalized. |
| | $c_{\mathrm{corr}}$ | 0.5 | Maximum score assigned to the correction sub-signal in engagement. |
| Slow loop | $\alpha_0$ | 1 | Symmetric Dirichlet prior concentration for each value of each adaptable dimension. |
| | $W$ | 20 | Number of recent sessions retained in the sliding window for prior updates. |
| | $\eta_{\mathrm{beh}}$ | 0.6 | EQS threshold for selecting Tier 1 positive examples. |
| | $\beta$ | 0.6 | Weight on EQS when combining EQS with subjective feedback for Tier 2 example selection. |
| | $(\lambda_{\mathrm{Tier1}}, \lambda_{\mathrm{Tier2}})$ | $(1.0, 0.5)$ | Fractional observation weights for Tier 1 and Tier 2 examples. |
| | $\alpha_{\max}$ | 0.8 | Maximum influence of the learned prior during prior blending. |
| EQS weight adaptation | $\eta$ | 0.05 | Step size for adapting EQS component weights across sessions. |
| | $(w_{\min}, w_{\max})$ | $(0.1, 0.6)$ | Bounds used when projecting adaptive EQS weights back onto the simplex. |

## A    Hyperparameters

Table 2 lists the remaining hyperparameters used by the scoring, feedback, and generation procedures.

## B    Instruction-Space Details

We specify the dimensions of the unified instruction schema, or UIO. The content axis is encoded in *PolicyJSON* (Table 3), and the relational axis is encoded in *PersonaJSON* (Table 4).

### B.1    Policy and Persona Resolution

Resolution proceeds through seven stages (Algorithm 1). The resolver first checks whether the state signal is usable, then maps affect to a delivery mode, adjusts the policy for cognitive load, blends slow-loop priors when available, applies fast-loop adjustments, and clips low-confidence extremes toward conservative defaults.

**Step 0: Data-quality gate.**  If `data_quality` is `UNAVAILABLE`, the resolver returns a safe-default UIO. Ordinal dimensions are set to their midpoints, structural count fields are left unset, the affective mode is set to A5 (Neutral), and confidence is set to 0.0. This gives the system a defined fallback when no reliable state signal is available.

Table 3: Content-axis dimensions.

| Group | Dimension | Type | Values |
|---|---|---|---|
| Structure | number_of_sentences | integer | — |
| Structure | number_of_paragraphs | integer | — |
| Language | reading_level | ordinal (4) | elementary, secondary, tertiary, specialist |
| Language | sentence_complexity | ordinal (3) | low, medium, high |
| Language | technicality | ordinal (3) | non-technical, semi-technical, technical |
| Explanation | explanation_depth | ordinal (3) | shallow, standard, deep |
| Explanation | information_density | ordinal (3) | low, medium, high |
| Explanation | examples_frequency | ordinal (3) | none, few, many |
| Interaction | number_of_questions | integer | — |
| Interaction | question_types | multi-select (6) | clarifying, diagnostic, probing, application, reflective, comparative |
| Interaction | choice_complexity | enum (3) | binary, multi-option, open-ended |

Table 4: Relational-axis dimensions.

| Group | Dimension | Type | Values |
|---|---|---|---|
| Needs | relational_need | enum (6) | connection, validation, being_heard, affirmation, comfort, none |
| Needs | relational_priority | ordinal (3) | primary, secondary, background |
| Needs | task_need | enum (7) | information, problem_solving, creation, learning, decision_support, planning, emotional_processing |
| Needs | task_priority | ordinal (3) | primary, secondary, background |
| Style | role_stance | enum (8) | collaborative, authoritative, informational, creative_partner, socratic, facilitative, empathetic_listener, conversational_peer |
| Modulation | challenge_level | ordinal (3) | low, medium, high |
| Modulation | supportiveness | ordinal (3) | low, medium, high |
| Modulation | directiveness | ordinal (3) | low, medium, high |
| Modulation | warmth | ordinal (3) | low, medium, high |
| Modulation | error_tolerance | ordinal (3) | low, medium, high |
| Modulation | emotional_attunement | ordinal (3) | low, medium, high |
| Modulation | interactivity | ordinal (3) | low, medium, high |

**Step 1: Confidence gate.** The resolver treats affective and cognitive confidence separately. Affective confidence is limited by the weaker of valence and arousal, while cognitive confidence follows the load estimate. Low affective confidence forces the neutral affective mode A5; low cognitive confidence sets load to MEDIUM. Joint confidence is used only for conservative clipping, so uncertainty in any state signal moves the final UIO toward less extreme delivery settings.

**Step 2: Mode classification.** Following Russell's circumplex (Russell, 1980), we map categorical valence and arousal to one of nine affective modes (Figure 3). If one affective axis is substantially less confident than the other, that axis is set to neutral before lookup. If overall affective confidence is low, the resolver uses A5.

**Step 3: Persona and base policy.** Each affective mode maps to a complete persona and base policy (Table 5). Negative valence modes use an empathetic_listener stance with high supportiveness, emotional attunement, and error tolerance. Neutral mode A5 uses an informational stance with minimal relational marking. High-arousal positive modes increase challenge and interactivity. The task_need field is initialized to information and is revised by the reasoning call.

Table 5: Affective modes and default persona presets.

| Mode | Label | Role stance | Relational need | Challenge | Supportiveness | Attunement |
|------|-------|-------------|-----------------|-----------|----------------|------------|
| A1 | Distressed | empathetic_listener | comfort (primary) | L | H | H |
| A2 | Alert | collaborative | none (background) | M | M | M |
| A3 | Energized | creative_partner | none (background) | H | M | M |
| A4 | Disengaged | empathetic_listener | being_heard (primary) | L | H | H |
| A5 | Neutral | informational | none (background) | M | L | L |
| A6 | Content | conversational_peer | none (background) | M | M | M |
| A7 | Drained | empathetic_listener | comfort (primary) | L | H | H |
| A8 | Attentive | informational | none (background) | M | M | M |
| A9 | Focused | socratic | none (background) | M | M | M |

**Step 4: Load modulation.** Cognitive load adjusts the base policy. Under `HIGH` load, the resolver lowers challenge and interactivity and tightens structural constraints. Under `LOW` load, it allows deeper and more interactive responses. When load confidence is low, the resolver uses `MEDIUM` to avoid overreacting to uncertain signals.

**Step 5: Slow-loop prior blending.** When slow-loop priors are available, the resolver blends the rule-based value with the learned user prior using Equation 5. Dimensions with little evidence remain close to the rule-based default, whereas dimensions with repeated positive evidence shift toward the learned user prior.

**Step 6: Fast-loop adjustment.** If the fast loop recommends adaptation, the resolver maps the recommendation to UIO dimensions with a fixed lexicon and applies a single ordinal step in the requested direction. If the failure is marked as an execution failure, the UIO is not changed; the next generation is instead prompted to satisfy the same UIO more faithfully.

**Step 7: Conservative bounds.** The final stage clips extreme ordinal values when confidence is low or data quality is degraded. Clipping moves each affected dimension one step toward the midpoint and records the change in the resolver audit trail.

## C  Response Generation

**Call 1: Reasoning.** The reasoning call returns JSON with three fields: the inferred task, the inferred relational need, and a fit assessment for the resolved policy and persona. Task inference assigns one of seven task categories (information, problem solving, creation, learning, decision support, planning, or emotional processing) and a confidence score. Relational read estimates the user's relational need and confidence. Fit assessment rates policy and persona fit as `good`, `partial`, or `poor`. On parse or completion failure, we use a default block with zero confidence, `poor` fit, and `used_fallback=true`. We run this call at temperature 0.0 in JSON mode with a 500-token cap.

**Call 2: Generation.** The generation call produces the user-facing response under a fixed priority order: safety constraints first, UIO constraints second, and default model behavior only where the UIO is silent. The prompt separates constraints into quantitative, language, explanation, and interaction groups, since these are evaluated by exact counts, distributional features, or semantic judgment. It also prevents the model from exposing internal objects such as the UIO, ReasoningBlock, or resolver state. We run this call at temperature 0.7 in free-text mode with a 700-token cap.

Figures 10 and 11 show the prompts used for both calls. Placeholders in braces, such as `{unified_instruction_object_json}`, are replaced at call time with serialized JSON values.

```
System Prompt — Call 1: Reasoning

You are operating under a state-conditioned interaction framework.
UNIFIED INSTRUCTION OBJECT:
{unified_instruction_object_json}
Before generating a user-facing response, you must complete a structured reasoning step.
Purpose:
- Interpret the user's underlying task.
- Classify the task need into one of the defined categories.
- Assess relational needs.
- Evaluate whether the resolved persona and policy fit this moment.
You must respond ONLY with a valid JSON object.
REQUIRED JSON SCHEMA:
{
  "task": {
    "inferred":   <string, exactly one sentence>,
    "confidence": <float, 0.0-1.0>,
    "evidence":   <string, exactly one sentence>,
    "task_need":  <enum: information | problem_solving | creation | learning
                  | decision_support | planning | emotional_processing>
  },
  "relational_read": {
    "assessment": <string, exactly one sentence>,
    "confidence": <float, 0.0-1.0>
  },
  "fit_assessment": {
    "persona_fit": <"good" | "partial" | "poor">,
    "policy_fit":  <"good" | "partial" | "poor">,
    "notes":       <string | null; null if both "good", else one sentence>
  }
}
Output only the JSON object.
```

Figure 10: Reasoning-call prompt.

# D  Subjective Component of EQS

We measure the subjective component of EQS with ten user-feedback items: eight 5-point Likert items and two free-text retrospective items. Table 6 lists the full inventory. Only Likert items contribute to $S_t$; free-text items are used for qualitative analysis.

Table 6: Subjective feedback items used for EQS and retrospective analysis.

| Field | User judgment | Collection point |
|---|---|---|
| overall_satisfaction | overall response satisfaction | scheduled |
| state_match | fit to current state | scheduled |
| tone_appropriateness | tone fit for the situation | scheduled |
| effort_required | effort required to use the response | scheduled |
| clarity | response clarity | scheduled |
| emotional_tone_match | fit to current emotional state | scheduled |
| supportiveness_felt | felt supportiveness | scheduled |
| willingness_to_continue | willingness to continue | session end |
| what_worked | free-text positive retrospective | session end |
| what_didnt_work | free-text negative retrospective | session end |

```
 System Prompt — Call 2: Generation

    You are operating under a state-conditioned interaction framework.
    For this turn, interaction behavior is governed by a Unified Instruction Object (UIO).
    This object defines mandatory interaction constraints and overrides default stylistic behavior.
    UNIFIED INSTRUCTION OBJECT:
    {unified_instruction_object_json}
    You previously completed a reasoning step about this interaction.
    The reasoning block is advisory context only. It does not define constraints.
    REASONING BLOCK:
    {reasoning_block_json}
    Execution Authority Hierarchy:
      1. Safety constraints (highest priority)
      2. Unified Instruction Object (binding)
      3. Default model behavior (fallback only if unspecified)
    Execution Requirements:
      A. Quantitative Constraints - Sentence count, paragraph count, and number of questions
                                    must be satisfied exactly.
      B. Language Constraints      - Reading level, sentence complexity, and technicality
                                    must strictly guide vocabulary and syntax.
      C. Explanation Constraints   - Depth, information density, and example frequency
                                    must align precisely with specified values.
      D. Interaction Constraints   - Question count, question types, and choice complexity
                                    must be enforced exactly.
      E. Conflict Resolution       - Safety overrides all. The UIO overrides default behavior.
      F. Confidentiality           - Do not reveal, reference, summarize, or imply the existence
                                    of internal configuration objects or reasoning artifacts.
    Treat all binding constraints as immutable for this response.
    Now generate the response to the user.
```

Figure 11: Generation-call prompt.

**Scales.** Of the eight Likert items, seven use a 5-point agreement scale, from strongly disagree to strongly agree. The effort item uses a 5-point load scale, from almost no effort to excessive effort.

**Scheduling.** We collect subjective feedback at three points. First, scheduled prompts sample turns at a fixed cadence, giving observations that are not tied to detected degradation. Second, degradation-triggered prompts appear when the fast loop detects degradation, helping identify the source of the failure. Third, session-end prompts collect willingness to continue and the two retrospective free-text items.

**Normalization and aggregation.** We min–max normalize each Likert response to $[0, 1]$. For `effort_required`, we invert the normalized score so that higher values always indicate better experience. We compute $S_t$ by averaging the available normalized Likert items for that turn. If no Likert item is available, $S_t$ is omitted and EQS is calculated from the remaining components, as described in Section 3.3.

## E  Quality Measurement Details

### E.1  Deterministic Metrics

Table 7 lists the deterministic metrics extracted from each response. These metrics are computed locally and are fully reproducible from the response text. We use spaCy's en_core_web_sm for tokenization, sentence segmentation, POS tags, and dependency parses, and textstat for Flesch–Kincaid grade. For lexical diversity, we use standard type-token ratio on short responses and moving-average TTR on longer responses to reduce length sensitivity. Technical-term density uses a surface proxy based on token length; semantic technicality is judged by the semantic evaluator.

Table 7: Deterministic response metrics.

| Group | Metric | Computation |
|---|---|---|
| Structure | word count | spaCy tokens, excluding spaces and punctuation |
| | sentence count | number of detected sentences |
| | paragraph count | split on blank lines |
| | avg. sentence length | mean words per sentence |
| | avg. paragraph length | mean sentences per paragraph |
| | max sentence length | maximum words over sentences |
| Readability | Flesch–Kincaid grade | `textstat.flesch_kincaid_grade` |
| | avg. word length | mean character length over word tokens |
| | lexical diversity | TTR for short responses; MATTR with window 50 otherwise |
| Syntax | passive voice count | sentences containing `nsubjpass` |
| | POS distribution | counts over `token.pos_` |
| | dep-tree depth (mean) | mean token depth to dependency root |
| | dep-tree depth (max) | maximum token depth to dependency root |
| | subordinate clauses | counts of `advcl`, `relcl`, `ccomp`, and `acl` |
| | coordination count | count of coordinating conjunction dependencies |
| Lexical | technical terms count | alphabetic tokens of length $\geq 8$ |
| | question count | sentences ending in ? |
| | imperative form count | root verb without an explicit subject child |
| | modal verb count | tokens tagged `MD` |
| | pronoun distribution | counts over pronoun tokens |
| | exclamation count | count of ! tokens |

## E.2 Semantic Feedback Engine

**Call settings.** We run the SFE as a separate stateless LLM call at temperature 0.0 in JSON output mode with a 1,500-token cap. In our implementation, it uses the same base model as the generation call, but it receives a separate evaluator prompt shown in Figure 12. The input contains the user message, generated response, resolved UIO, the ReasoningBlock from Call 1, the user-state snapshot, and the last $K{=}10$ turns of conversation history.

**Output schema.** The SFE returns one JSON object with three parts. First, `safety_brake` reports whether crisis language is detected, its severity, supporting indicators, and a recommended safety action. Second, `semantic_compliance` reports an `overall_fit` score in $[0, 1]$ and per-dimension scores for `reading_level`, `challenge_level`, `interactivity`, `tone`, `encouragement`, and `detail_level`. Each dimension includes a short rationale. This field also records whether a structural policy override occurred and whether it was contextually justified. Third, `adaptation_signal` reports whether the next turn should adapt, up to two target dimensions, the recommended direction (`increase`, `decrease`, or `maintain`), the requested ordinal step, and a brief rationale. It also includes an engagement trend label: `disengaged`, `passive`, `engaged`, or `highly_engaged`.

**Parsing and recovery.** We parse SFE output with a three-step recovery procedure. We first apply direct schema validation. If that fails, we parse the response with `json.loads` and validate again. If parsing still fails, we scan for the outermost JSON object and retry validation on that substring. The schema also accepts minor aliases, such as `dimensions` for `per_dimension`. If all recovery steps fail, we fall back to deterministic structural compliance checks and log the raw response in the session audit trail.

## F    Experimental Process

Our simulator produces two outputs for each turn: behavioral signals for EQS and subjective ratings for the feedback-enabled conditions. To keep the ratings grounded in human data, we calibrate the rating distribution against PRISM human feedback. PRISM ratings are used for calibration only; they are not used to score our generated responses directly.

**Simulated perception.**    For each generated response, the simulator computes five latent perception signals. *Compliance* is the semantic evaluator's overall semantic-fit score. *Supportiveness* measures validation, empathy, and encouragement markers. *Clarity* combines readability with penalties for overly long sentences and excessive sentence count. *Effort* estimates processing burden from Flesch–Kincaid grade, maximum sentence length, response length, cognitive load, and arousal. *Tone fit* weights supportiveness by the user's current affective state.

Two state-dependent assumptions shape the final ratings.  First, distressed states increase sensitivity to relevance and state match, so compliance errors have larger cost in more distressed states.  Second, supportiveness is asymmetric under distress: high supportiveness improves tone fit, but low supportiveness is penalized more strongly. We also add a deterministic per-subject bias in the range $\pm 0.09$ to model stable differences in rating style. The resulting scores are mapped to the 1–5 Likert scale, with `effort_required` inverted before scoring.

**Calibration.**    We calibrate the simulator against 774 PRISM per-turn human ratings from the 116 selected prompts. The main calibration target is the affective-tone gradient in mean satisfaction: crisis-sensitive < negative non-crisis < positive < neutral (4.31, 4.64, 4.74, and 4.90). The simulator is constrained to reproduce this ordering. We do not force exact equality in absolute ratings because our systems generate different responses from the PRISM models.

**Feedback schedule.**    Turn-level subjective feedback is available only in feedback-enabled sessions. It is collected every third turn and after two consecutive degrading turns. Session-end feedback is collected once per feedback-enabled session and includes `willingness_to_continue`. Session-end feedback also includes a binary `preference_vs_alternative` judgment, which is set when normalized willingness to continue is at least 0.6.

In Run D, each subject has two feedback-enabled sessions out of eight. We use five fixed session-pair templates. The held-out distressed subject S13 receives $(1, 5)$ to ensure that heightened state sensitivity appears early enough to be observed by the slow loop. The four regular subjects receive $(2, 6)$, $(3, 7)$, $(4, 8)$, and $(1, 6)$ in sorted subject order.

## G    Subject Selection

DEAP provides 40 trials for each of 32 subjects, with self-reported valence and arousal on 1–9 scales. We discretize both dimensions using the Tripathi thresholds, with cuts at 4 and 6. The same thresholds are used for subject selection and experiment execution.

**Filtering.**    We exclude subjects with restricted affective range: $\mathrm{SD}(V) < 1.5$ or $\mathrm{SD}(A) < 1.5$. These subjects cannot reliably populate the extreme cells of the valence–arousal grid. This filter leaves 20 subjects.

**Greedy selection.**    From the filtered pool, we select eight subjects by greedy coverage of the six target affective modes.  Candidates are ranked by grid occupancy, target-mode breadth and depth, and lower absolute valence–arousal correlation. At each step, we add the subject that contributes the most uncovered target modes, breaking ties by progress toward an 8-trial-per-mode floor. The selected set reaches at least eight trials for every target mode in the union. The shallowest target mode is $A_8$, with 27 trials. Subject S04 is retained despite a high valence–arousal correlation because it supplies unusually strong coverage of $A_4$.

Table 8: Simulated subjective-feedback fields.

| Field | Simulation rule |
|---|---|
| *Turn-level items* | |
| overall_satisfaction | Weighted mean of compliance, state match, tone fit, inverse effort, and clarity, plus subject bias |
| state_match | Supportiveness, clarity, and compliance, reweighted by valence and arousal |
| tone_appropriateness | Supportiveness and compliance with a state-dependent adjustment under distress |
| effort_required | Processing cost from readability, maximum sentence length, response length, load, and arousal |
| clarity | Readability score with penalties for long sentences and excessive sentence count |
| emotional_tone_match | Tone-fit signal with stronger state weighting |
| supportiveness_felt | Density of validation, empathy, and encouragement markers |
| *Session-end items* | |
| overall_satisfaction | Session mean of turn-level satisfaction, or mean compliance, clarity, and supportiveness if turn ratings are absent |
| clarity | Session mean of clarity, plus subject bias |
| supportiveness_felt | Session mean of supportiveness, plus subject bias |
| willingness_to_continue | Mean of session satisfaction, supportiveness, and compliance, plus subject bias |
| preference_vs_alternative | Binary threshold on normalized willingness to continue ($\geq 0.6$) |
| what_worked | Reserved for human collection |
| what_didnt_work | Reserved for human collection |

**Representative trials.** For each subject–mode pair, we select the trial with the largest joint subject-normalized deviation from that subject's valence and arousal baselines. This gives the most salient instance of the mode for that subject. Ties are broken by trial id.

**Run D subject pool.** Run D uses five subjects: S22, S04, S05, S12, and the held-out distressed subject S13. S13 ranks tenth in the filtered pool, passes the correlation criterion, and contributes 10 trials of $A_1$. We exclude it from Runs A/B so that the single-turn controllability analysis is not skewed by a persistent distress profile.

# H   Prompt Selection

PRISM contains 8,011 human–LLM conversations with per-turn human quality scores and end-of-conversation helpfulness ratings. We use only the user turns. PRISM model responses are never shown to our system.

**Filtering.** We apply two gates. First, we keep conversations with at least six user turns, matching our target session length. Second, we remove conversations in which more than 30% of follow-up turns depend on a specific prior model response, such as "you said" or "the first option." About 730 conversations pass these filters.

**Classification.** We classify each remaining conversation by affective tone and task type. Affective tone has four values: positive, negative_non_crisis, neutral, and crisis_sensitive. Task type has three values: information, problem_solving, and discussion. This yields twelve prompt categories. The classifier sees only the opening user turn and the first two follow-up user turns, with no model responses. We run it at temperature 0.0 with strict JSON output. The prompt appears in Figure 13.

**Ranking and selection.** Within each category, we rank conversations by a fixed weighted score: 0.30 turn-count fit, 0.25 mean PRISM quality, 0.20 self-containment, and 0.25 PRISM helpfulness. Missing quality or helpfulness values are set to 0.5. The turn-count component favors our target session length:

| turns | $< 5$ | 5 | 6–8 | 9 | $\geq 10$ |
|-------|-------|-----|-----|-----|-----------|
| score | 0.3 | 0.7 | 1.0 | 0.8 | 0.6 |

We retain the top ten conversations per category. Three `crisis_sensitive` categories contain fewer than ten eligible conversations after filtering, leaving 116 prompts. For S13 in Run D, acute crisis-sensitive prompts are hand-crafted under the same authorship constraints because PRISM excludes acute self-harm content.

**Assignment.** Runs A/B use a balanced-block assignment over the 288 trial cells. A fixed-seed category rotation reduces dependence between prompt category and cell index. For Run D, each subject receives one prompt per session across eight sessions, drawn from the nine non-safety categories. One category is dropped per subject, and the dropped category rotates across subjects. S13 uses a fixed sequence of four `crisis_sensitive` and four `negative_non_crisis` prompts, with mode fixed at $A_1$ and high cognitive load throughout.

**Stored fields.** Each selected prompt is stored with its PRISM id, classified tone and task, safety flag, up to eight user turns, and ranking metadata. Conversations longer than eight user turns are truncated and flagged.

# I Conditions and Runs

We compare three conditions: the full adaptive system ($C_0$), the conditioned-only ablation ($C_2$), and the unconditioned baseline ($C_4$). Table 9 summarizes the active components.

Table 9: Condition definitions.

| Condition | System prompt | UIO | Fast loop | Slow loop |
|-----------|:-------------:|:---:|:---------:|:---------:|
| $C_0$ | ✓ | ✓ | ✓ | ✓ |
| $C_2$ | ✓ | ✓ | — | — |
| $C_4$ | — | — | — | — |

**Run design.** We organize evaluation into four runs. Runs A and B are matched single-turn experiments over the same 288 subject–mode–load–prompt cells. Run A uses the $C_0$ pipeline in cold-start mode for each trial, so UIO conditioning is active but no cross-turn adaptation history is carried between trials. Run B uses $C_4$ on the same cells, isolating the effect of delivery conditioning.

Run D evaluates multi-session adaptation. Each subject completes eight sessions of 6–8 turns with a mode shift at turn 4. Conditions $C_0$ and $C_2$ are run back-to-back per session. To make the comparison matched, each $C_2$ response is also scored against the corresponding $C_0$ resolved UIO, so that differences reflect response realization rather than different target instructions. Run D* replays the Run D prompts and mode-shift schedule through the public ChatGPT and Claude interfaces to construct the multi-session $C_4$ arm; responses are scored post hoc by the SFE.

**Models.** Runs A/B use `gpt-5.4` and `claude-sonnet-4.6`. Within these runs, reasoning, generation, and SFE scoring use the same model family. For Run D and Run D*, the GPT condition uses `gpt-5.3-chat` for reasoning and generation because the multi-session $C_4$ arm is collected through the public ChatGPT interface. The Claude condition uses `claude-sonnet-4.6` for all generation-side calls. All Run D and Run D* SFE scoring uses `gpt-5.4`, keeping compliance judgments comparable across conditions.

Table 10: Evaluation runs.

| Run | Conditions | Unit | Design size | Calls/turn |
|---|---|---|---|---|
| A | $C_0$ | trial | 288 trials | 3 |
| B | $C_4$ | trial | 288 trials | 2 |
| D | $C_0$, $C_2$ | session | 40 sessions (259 turns) | 3 (+1 for $C_2$ rescore) |
| D$^*$ | $C_4$ | session | same schedule as D | 1 post-hoc SFE |

**Compute cost.** Table 11 reports token usage and cost for the complete evaluation. Costs are computed from the rate cards used at collection time: `gpt-5.4` at \$2.50/\$15.00 per million input/output tokens, `gpt-5.3-chat` at \$1.75/\$14.00, and `claude-sonnet-4.6` at \$3.00/\$15.00. The total evaluation cost is \$133.26.

Table 11: Token usage and cost by run, condition, and model.

| Run | Cond. | Model | $R$ | $G$ | $E$ | Total | Cost |
|---|---|---|---|---|---|---|---|
| A | $C_0$ | GPT | 244,958 | 239,633 | 697,751 | 1,182,342 | 10.35 |
| A | $C_0$ | Claude | 286,863 | 273,143 | 906,724 | 1,466,730 | 13.20 |
| B | $C_4$ | GPT | — | 91,328 | 691,090 | 782,418 | 6.85 |
| B | $C_4$ | Claude | — | 77,797 | 905,299 | 983,096 | 8.85 |
| D | $C_0$ | GPT | 353,456 | 340,253 | 703,040 | 1,396,749 | 11.61 |
| D | $C_0$ | Claude | 409,306 | 399,296 | 982,819 | 1,791,421 | 15.88 |
| D | $C_2$ | GPT | 349,555 | 334,598 | 698,886 | 1,383,039 | 11.50 |
| D | $C_2$ | Claude | 406,584 | 394,625 | 964,342 | 1,765,551 | 15.65 |
| D | $C_2$ rescore | GPT | — | — | 1,014,318 | 1,014,318 | 8.88 |
| D | $C_2$ rescore | Claude | — | — | 1,317,217 | 1,317,217 | 11.53 |
| D$^*$ | $C_4$ | GPT | — | — | 975,312 | 975,312 | 8.53 |
| D$^*$ | $C_4$ | Claude | — | — | 1,193,789 | 1,193,789 | 10.45 |
| | | | | | **Total** | **15,251,982** | **133.26** |

Here, $R$ denotes reasoning tokens, $G$ generation tokens, and $E$ SFE scoring tokens. In Run D and Run D$^*$, SFE tokens are charged to `gpt-5.4` regardless of the generation model.

## J  RQ1: Full Results

We report four analyses for RQ1: affective-mode effects, cognitive-load effects, UIO compliance, and cross-model robustness.

### J.1  Affective-Mode Effects

Table 12 reports the MANOVA results for the nine-feature delivery vector. Under $C_0$, affective mode has a large effect on the measured delivery features for both models. The same test under $C_4$ is much smaller, indicating that unconditioned models show weaker mode-correlated variation from the prompts themselves.

Table 12: Affective-mode MANOVA for the nine-feature delivery vector.

| Model | Condition | $N$ | Wilks' $\Lambda$ | $F$ | $p$ | $\eta_p^2$ |
|---|---|---|---|---|---|---|
| Claude | $C_0$ | 285 | 0.296 | 7.35 | $< 0.001$ | 0.194 |
| Claude | $C_4$ | 288 | 0.773 | 1.61 | 0.007 | 0.049 |
| GPT | $C_0$ | 288 | 0.291 | 7.80 | $< 0.001$ | 0.202 |
| GPT | $C_4$ | 288 | 0.692 | 2.22 | $< 0.001$ | 0.067 |

Table 13 decomposes the multivariate effect by feature. All nine features show significant affective-mode effects under $C_0$ on both models after Holm correction ($p_{\mathrm{Holm}} < .001$). In every case, the effect is larger under $C_0$ than under $C_4$.

Table 13: Per-feature affective-mode effects under $C_0$ and $C_4$.

| Feature | Claude $C_0$ $\eta^2$ | Claude $C_4$ $\eta^2$ | Ratio | GPT $C_0$ $\eta^2$ | GPT $C_4$ $\eta^2$ | Ratio |
|---|---|---|---|---|---|---|
| technical_term_density | 0.469 | 0.036 | 13.1× | 0.346 | 0.014 | 24.4× |
| flesch_kincaid_grade | 0.317 | 0.005 | 64.8× | 0.174 | 0.061 | 2.9× |
| word_count | 0.291 | 0.010 | 29.4× | 0.251 | 0.025 | 9.9× |
| lexical_diversity | 0.112 | 0.010 | 11.4× | 0.096 | 0.010 | 9.7× |
| validation_phrases | 0.220 | 0.090 | 2.5× | 0.075 | 0.005 | 14.7× |
| empathy_expressions | 0.147 | 0.088 | 1.7× | 0.104 | 0.006 | 17.2× |
| semantic_tone_score | 0.232 | 0.103 | 2.3× | 0.144 | 0.005 | 27.4× |
| semantic_interactivity_score | 0.159 | 0.018 | 9.0× | 0.218 | 0.005 | 41.6× |
| modal_verb_count | 0.073 | 0.004 | 18.5× | 0.104 | 0.095 | 1.1× |

These results indicate that the RQ1 effect is not driven only by response length. The largest changes appear on directly controlled structural and complexity features, such as technical density, reading level, and word count. Relational features also vary consistently, although with smaller effects.

## J.2 Cognitive-Load Effects

Table 14 repeats the MANOVA using cognitive load as the independent variable. The pattern mirrors the affective-mode analysis: $C_0$ produces substantially larger delivery shifts than $C_4$ on both models.

Table 14: Cognitive-load MANOVA for the nine-feature delivery vector.

| Model | Condition | $N$ | Wilks' $\Lambda$ | $F$ | $p$ | $\eta_p^2$ |
|---|---|---|---|---|---|---|
| Claude | $C_0$ | 285 | 0.387 | 16.08 | $< 0.001$ | 0.345 |
| Claude | $C_4$ | 288 | 0.866 | 2.30 | 0.002 | 0.069 |
| GPT | $C_0$ | 288 | 0.425 | 13.98 | $< 0.001$ | 0.312 |
| GPT | $C_4$ | 288 | 0.851 | 2.59 | $< 0.001$ | 0.077 |

## J.3 UIO Compliance

We evaluate UIO compliance only for $C_0$, because $C_4$ has no target UIO. Table 15 reports structural compliance and semantic fit. Direct structural constraints have the highest compliance: paragraph count, sentence count, and reading level all exceed 92% on both models. Technicality is weaker, showing that lexical-distributional control is harder than count or reading-level control.

The semantic scores show the same difficulty ordering on both models: encouragement is hardest, followed by tone, interactivity, and challenge_level. This pattern suggests that, in our setting, lower scores are associated with the semantic difficulty of realizing persona constraints rather than with a model-specific failure.

## J.4 Cross-Model Robustness

The controllability pattern is consistent across models. Per-feature $C_0$ effect sizes correlate strongly between Claude and GPT (Pearson $r=0.80$; Spearman $\rho=0.67$), and all nine features are significant under $C_0$ for both models. The MANOVA effects are also closely matched: affective-mode $\eta_p^2$ is 0.194 for Claude and 0.202 for GPT, while cognitive-load $\eta_p^2$ is 0.345 for Claude and 0.312 for GPT. Across all nine features, the affective-mode effect is larger under $C_0$ than under $C_4$ on both models.

Table 15: UIO compliance under $C_0$.

| Metric | Scale | Claude | GPT |
|---|---|---|---|
| paragraph_count | compliance rate | 98.6% | 100.0% |
| sentence_count | compliance rate | 92.6% | 95.8% |
| reading_level | compliance rate | 92.1% | 92.1% |
| technicality | compliance rate | 50.0% | 56.0% |
| overall_fit | SFE score | 0.735 | 0.839 |
| challenge_level | SFE score | 0.797 | 0.879 |
| interactivity | SFE score | 0.779 | 0.856 |
| tone | SFE score | 0.720 | 0.842 |
| encouragement | SFE score | 0.633 | 0.765 |

## K  RQ2: Full Results

We evaluate eight session-level outcomes for RQ2: EQS, SFE overall fit, and six SFE dimensions (reading_level, challenge_level, interactivity, tone, encouragement, and detail_level). Each outcome is averaged within session before comparison. Paired tests match the same subject and session index across conditions, giving $N{=}40$ session pairs per model and $N{=}80$ when pooled.

### K.1  Pooled Pairwise Comparison

Table 16 reports the pooled comparison among the full system ($C_0$), the conditioned-only ablation ($C_2$), and the unconditioned baseline ($C_4$). In the pooled means, the ordering is consistent: $C_0 > C_2 > C_4$ on all eight outcomes.

Table 16: Pooled RQ2 pairwise comparisons.

| Outcome | $C_0$ | $C_2$ | $C_4$ | $C_0{-}C_4$ $d$ | $C_0{-}C_4$ $p$ | $C_0{-}C_2$ $d$ | $C_0{-}C_2$ $p$ | $C_2{-}C_4$ $d$ | $C_2{-}C_4$ $p$ |
|---|---|---|---|---|---|---|---|---|---|
| eqs | 0.719 | 0.708 | 0.592 | 1.209 | < 0.001 | 0.474 | 0.001 | 1.040 | < 0.001 |
| overall_fit | 0.798 | 0.775 | 0.566 | 1.255 | < 0.001 | 0.653 | < 0.001 | 1.080 | < 0.001 |
| sfe_reading_level | 0.807 | 0.792 | 0.636 | 0.865 | < 0.001 | 0.279 | 0.041 | 0.712 | < 0.001 |
| sfe_challenge_level | 0.845 | 0.833 | 0.700 | 0.944 | < 0.001 | 0.394 | < 0.001 | 0.863 | < 0.001 |
| sfe_interactivity | 0.800 | 0.758 | 0.546 | 1.151 | < 0.001 | 0.628 | < 0.001 | 0.941 | < 0.001 |
| sfe_tone | 0.804 | 0.784 | 0.708 | 0.613 | < 0.001 | 0.433 | 0.003 | 0.444 | < 0.001 |
| sfe_encouragement | 0.728 | 0.702 | 0.629 | 0.637 | < 0.001 | 0.516 | < 0.001 | 0.424 | < 0.001 |
| sfe_detail_level | 0.819 | 0.800 | 0.595 | 1.144 | < 0.001 | 0.427 | 0.001 | 1.005 | < 0.001 |

*Note.* Values in the $C_0$, $C_2$, and $C_4$ columns are session-level means pooled across both models. Effect sizes are paired-design Cohen's $d_z$. Reported $p$-values are Holm-corrected within each pairwise family.

The full system has higher session-level means than $C_4$ on all eight outcomes, with large effects for EQS, SFE overall fit, interactivity, and detail level. The conditioned-only ablation also has higher means than $C_4$ throughout, indicating that state-conditioned delivery explains most of the observed gap. The $C_0{-}C_2$ contrast shows a smaller but consistent additional gain, especially on overall fit, interactivity, tone, encouragement, and detail level.

### K.2  Attribution Decomposition

Table 17 decomposes the total $C_0{-}C_4$ gap into the contribution of state conditioning ($C_2{-}C_4$) and the added contribution of feedback loops ($C_0{-}C_2$).

Conditioning explains most of the total $C_0{-}C_4$ gap, accounting for 74–91% across outcomes. Feedback loops account for the remaining 9–26%. The loop share is largest on relational dimensions, especially encouragement and tone.

Table 17: Attribution decomposition of the pooled $C_0 - C_4$ gain.

| Outcome | Total gap | Conditioning gap | Loop gap | Conditioning share | Loop share |
|---|---|---|---|---|---|
| overall_fit | 0.232 | 0.209 | 0.024 | 90% | 10% |
| sfe_tone | 0.097 | 0.076 | 0.020 | 79% | 21% |
| sfe_encouragement | 0.098 | 0.073 | 0.025 | 74% | 26% |
| sfe_interactivity | 0.254 | 0.212 | 0.042 | 83% | 17% |
| sfe_challenge_level | 0.145 | 0.132 | 0.013 | 91% | 9% |
| sfe_reading_level | 0.171 | 0.156 | 0.015 | 91% | 9% |
| sfe_detail_level | 0.225 | 0.205 | 0.020 | 91% | 9% |

### K.3 Sensitivity Analysis

Table 18 reports the sensitivity checks for the primary $C_0 > C_4$ effect on SFE `overall_fit`. The effect remains positive and significant under every perturbation.

Table 18: Sensitivity checks for the $C_0 > C_4$ effect on SFE overall fit.

| Analysis | $N$ | $\Delta$ | $d$ | 95% CI for $d$ | $p$ |
|---|---|---|---|---|---|
| Primary pooled analysis | 80 | +0.232 | 1.255 | [0.966, 1.542] | $< 0.001$ |
| Excluding $S_{13}$ | 64 | +0.228 | 1.185 | [0.885, 1.465] | $< 0.001$ |
| GPT only | 40 | +0.110 | 1.098 | [0.503, 1.497] | $< 0.001$ |
| Claude only | 40 | +0.355 | 2.092 | [1.514, 2.966] | $< 0.001$ |
| Bonferroni correction | 80 | +0.232 | 1.255 | [0.966, 1.542] | $< 0.001$ |
| SFE fallback turns treated as missing | 80 | +0.232 | 1.255 | [0.966, 1.542] | $< 0.001$ |
| Subjective-feedback sessions excluded | 80 | +0.232 | 1.255 | [0.966, 1.542] | $< 0.001$ |

The effect remains after excluding the distressed subject, restricting analysis to either model, changing the multiple-comparison correction, treating fallback-scored turns as missing, and excluding sessions that include subjective feedback. In all checks, the effect remains large ($d > 1.0$) and statistically significant.

## L    RQ3: Full Results

We report RQ3 results separately for the fast loop, which acts within a session, and the slow loop, which accumulates evidence across sessions.

### L.1    Fast-Loop Targeting and Recovery

Table 19 reports fast-loop targeting and next-turn recovery under $C_0$. For each adjustment, we map the targeted UIO dimension to the corresponding SFE feature and measure the change in that feature on the following turn. We also report the weakest-dimension targeting rate, defined as the fraction of adjustments that targeted the lowest-scoring SFE dimension.

The fast loop targets the weakest SFE dimension in 96.2% of Claude adjustments and 98.5% of GPT adjustments. Mean next-turn change is positive for all listed target dimensions. The largest gains occur on structural and cognitive dimensions, especially `reading_level` and `challenge_level`. Relational targets show smaller positive mean changes, especially `supportiveness` and `emotional_attunement`. We omit `information_density` from the table because it appears only three times, all on GPT, which is too sparse for a stable estimate.

Table 19: Fast-loop targeting and next-turn recovery under $C_0$.

| Target dimension | Matched SFE feature | Claude $N$ | Claude $\Delta$ | Claude recovery | GPT $N$ | GPT $\Delta$ | GPT recovery |
|---|---|---|---|---|---|---|---|
| reading_level | sfe_reading_level | 32 | +0.135 | 75.0% | 88 | +0.106 | 69.3% |
| explanation_depth | sfe_detail_level | 88 | +0.033 | 62.5% | 54 | +0.079 | 74.1% |
| challenge_level | sfe_challenge_level | 33 | +0.060 | 57.6% | 26 | +0.118 | 80.8% |
| supportiveness | sfe_encouragement | 181 | +0.027 | 48.1% | 158 | +0.035 | 52.5% |
| emotional_attunement | sfe_encouragement | 115 | +0.057 | 53.0% | 57 | +0.064 | 61.4% |
| interactivity | sfe_interactivity | 89 | +0.051 | 61.8% | 91 | +0.059 | 64.8% |
| Weakest-dimension targeting rate | | 96.2% | [92.7%, 98.1%] | | 98.5% | [95.7%, 99.5%] | |

## L.2 Slow-Loop Trajectory

Table 20 summarizes the three trajectory measures used in the main text: the blending factor $\alpha_{\mathrm{eff}}$, JS divergence from the uniform prior, and Hamming distance between the conditioning-only UIO and the prior-blended UIO on slow-only turns.

Table 20: Slow-loop trajectory across eight sessions under $C_0$.

| Measure | Model | Session 1 | Session 8 | Linear trend | $p$ |
|---|---|---|---|---|---|
| $\alpha_{\mathrm{eff}}$ | Claude | 0.430 | 0.799 | +0.050 | < 0.001 |
| $\alpha_{\mathrm{eff}}$ | GPT | 0.413 | 0.799 | +0.049 | < 0.001 |
| JS divergence from uniform prior | Claude | 0.000 | 0.097 | +0.012 | < 0.001 |
| JS divergence from uniform prior | GPT | 0.000 | 0.104 | +0.013 | < 0.001 |
| Hamming distance from conditioning-only UIO | Claude | 0.000 | 2.600 | +0.541 | < 0.001 |
| Hamming distance from conditioning-only UIO | GPT | 0.000 | 2.400 | +0.543 | < 0.001 |

All three measures increase significantly across sessions on both models. The blending factor approaches the design cap of 0.80 by session 8. JS divergence grows from the uniform prior, indicating that the slow loop accumulates subject-specific evidence. The Hamming distance also increases, indicating that the learned prior affects the final UIO rather than remaining only a latent statistic.

## L.3 Between-Subject Prior Divergence

Table 21 reports terminal between-subject divergence at session 8. We compare the modal values of the learned priors across subjects over the 14 slow-loop dimensions.

Table 21: Terminal between-subject divergence of learned priors.

| Model | Subjects | Pairs | Mean Hamming distance | Max possible |
|---|---|---|---|---|
| Claude | 5 | 10 | 5.0 | 14 |
| GPT | 5 | 10 | 4.4 | 14 |

By the final session, learned priors differ across subjects by an average of 5.0 dimensions on Claude and 4.4 dimensions on GPT. Since the uniform prior assigns the same modal category to every subject, its between-subject Hamming distance is zero. The observed divergence indicates that the slow-loop priors become subject-specific rather than remaining identical across subjects.

## L.4 Dimension-Level Prior Divergence

Table 22 shows where terminal between-subject divergence concentrates. Values are the percentage of subject pairs whose learned prior modal values differ on each dimension.

Table 22: Terminal between-subject disagreement by dimension.

| Dimension | Claude | GPT |
|---|---|---|
| challenge_level | 80% | 80% |
| explanation_depth | 60% | 40% |
| emotional_attunement | 60% | 40% |
| reading_level | 40% | 60% |
| sentence_complexity | 40% | 60% |
| information_density | 40% | 40% |
| interactivity | 40% | 40% |
| technicality | 0% | 0% |
| examples_frequency | 0% | 0% |
| choice_complexity | 0% | 0% |

Divergence is concentrated in challenge_level, explanation_depth, and emotional_attunement, while technicality, examples_frequency, and choice_complexity remain aligned across subjects. This indicates that the slow loop changes some delivery dimensions more than others, rather than shifting all dimensions uniformly.

---

**System Prompt — Semantic Feedback Engine**

You are the Semantic Feedback Engine for an adaptive conversational system.
You receive the full context of a conversational turn and produce three
assessments.

Return ONLY valid json matching the TARGET_SCHEMA. No markdown, no commentary,
no extra keys.

ASSESSMENT 1 - SAFETY BRAKE
Evaluate whether the conversation content combined with physiological state
warrants crisis intervention. Activate the safety brake when you detect:
  - Explicit self-harm or suicidal ideation
  - Escalating violence toward self or others
  - Signs of acute psychological crisis (dissociation, panic, hopelessness
    progression)
Consider the physiological state: if the user's body signals confirm distress
(negative valence, high arousal), this STRENGTHENS the case for activation.
If physiological signals are calm despite concerning language, assess whether
this might be academic/hypothetical discussion.
Severity levels:
  - none: Normal conversation
  - low: Mild concerning language, monitor
  - medium: Concerning patterns emerging, increase support
  - high: Active crisis indicators, recommend safe mode
  - critical: Immediate danger signals, require safe mode

ASSESSMENT 2 - SEMANTIC COMPLIANCE
Evaluate whether the response matched the SPIRIT of the policy and persona,
not just structural targets. A response that exceeds word count but is
perfectly warm, simple, and supportive during a crisis should score high.
Judge intent alignment, not metric compliance.
Dimensions to evaluate: reading_level, challenge_level, interactivity, tone,
encouragement, detail_level. For each, provide a 0-1 score and one-sentence
rationale.
If the response overrode structural policy constraints, assess whether that
override was justified by the conversational context (e.g., crisis
intervention requiring more content).

ASSESSMENT 3 - ADAPTATION DIRECTION
First inspect the semantic-compliance scores you assigned. If should_adapt
is true, the adaptations must target the lowest-scoring underperforming
dimensions rather than defaulting to maintain-only recommendations.
Recommend 1-2 specific dimension changes for the next turn based on BOTH:
  - The physiological state (HSI snapshot: valence, arousal, cognitive load)
  - The conversational trajectory (user engagement, follow-up patterns,
    topical continuity)
Neither signal alone is sufficient. A stressed user asking follow-up
questions may need maintained support but increased depth. A calm user
giving short responses may need increased engagement.
Use these adaptation dimensions when applicable: reading_level,
challenge_level, interactivity, tone, encouragement, detail_level,
information_density, explanation_depth.
When reading_level, challenge_level, interactivity, tone, encouragement, or
detail_level scores below 0.80, prefer at least one non-maintain
recommendation for the weakest 1-2 dimensions unless safety requires
stability.
Map low interactivity to more turn-taking or questioning, low encouragement
to more support, low tone to warmer attunement, and low detail_level to more
explanation detail.
Do not emit placeholder maintain-only adaptations when a clear low-scoring
dimension exists.
Assess user engagement: score 0-1, list observed signals, classify trend.

Figure 12: SFE call prompt.

**System Prompt — Conversation Classifier**

```
You are a conversation classifier. Given the opening turns of a human-LLM
conversation, classify it on two dimensions.
Affective Tone (the user's emotional state):
- positive: enthusiasm, gratitude, excitement, joy, curiosity with positive energy
- negative_non_crisis: frustration, sadness, dissatisfaction, worry, complaint
  (no immediate safety risk)
- neutral: balanced, factual, no strong emotional signal
- crisis_sensitive: distress involving mental health, safety, abuse, self-harm,
  grief, trauma, burnout, panic
Cognitive Task Type (what the user is trying to accomplish):
- information: seeking facts, explanations, understanding, learning about a topic
- problem_solving: pursuing a practical goal, making a decision, getting something
  done
- discussion: exploring opinions, debating values, discussing ethics or
  controversial topics, seeking dialogue rather than answers
Return ONLY the JSON classification. Do not explain.
```

Figure 13: PRISM prompt-classification prompt.

