# OpenReview forum: "How You Say It Matters: Personalizing LLM Responses via Dual Time-Scale Closed-Loop Adaptation"
_TMLR — Under review for TMLR_

### Review · Reviewer_Yfgh · 2026-06-24

**Summary Of Contributions:**

This paper studies personalization of response delivery rather than only response content. It introduces a structured instruction object for controlling properties such as tone, detail, warmth, and technicality, together with a fast loop for within-session correction and a slow loop for learning longer-term user preferences.

The framing is interesting, the system is clearly described, and the experiments show that the proposed instructions can reliably change response style. However, the evidence for improved user experience is less convincing because the evaluation mainly relies on simulated users and an LLM evaluator that is also part of the adaptation pipeline.

**Audience:**

Yes

**Audience Explanation:**

The paper should be interesting to researchers working on LLM personalization, conversational systems, and human–AI interaction, even though stronger user-grounded evaluation is still needed.

**Broader Impact Concerns:**

The paper should discuss consent, privacy, data retention, user correction and reset mechanisms, and safeguards against persuasive or manipulative use.

**Claims And Evidence:**

No

**Claims Explanation:**

The paper provides strong evidence that the UIO can control measurable response properties such as length, readability, and technicality.

However, the broader personalization claims are not fully supported. The main quality metrics depend heavily on the Semantic Feedback Engine, which sees the target UIO and is also used by the fast loop. This makes the evaluation somewhat circular: the system is instructed to follow a rubric and is then judged using the same rubric.

The interaction experiments also use simulated users. This is useful for controlled testing, but it does not show that real users would prefer the adapted responses or that the inferred user states are correct.

The evidence for the two feedback loops is also limited. The slow-loop results mainly show that the prior becomes less uniform over time, not that it predicts future user preferences better. The fast-loop recovery results lack a matched no-adjustment control, so some improvement may come from regression to the mean or differences between turns.

Finally, most of the gain appears to come from state conditioning rather than closed-loop adaptation, but the paper does not include separate fast-only and slow-only ablations.

**Requested Changes:**

1. Add a more independent evaluation, ideally blinded human judgments or an evaluator that does not see the target UIO or internal reasoning.
2. Include real-user evaluation, or clearly limit the claims to simulated interaction quality.
3. Add fast-only and slow-only ablations. For the fast loop, include a matched no-adjustment baseline. For the slow loop, test whether learned priors improve future held-out sessions.
4. Measure whether adaptation changes factual accuracy, task success, or the core information in the response.
5. Add simpler baselines, such as direct user-state prompting and static personalized style prompts.
6. Report inference cost and latency, since the framework requires several model calls per turn.
7. Test robustness to noisy or incorrect user-state estimates.

---

### Review · Reviewer_oaw8 · 2026-06-25

**Summary Of Contributions:**

The paper argues that LLM personalization research has focused almost entirely on what content to produce (retrieval, prompting, fine-tuning, RLHF), and has underemphasized how that content is delivered: its structure, tone, and relational stance. The authors make delivery an explicit personalization target and propose a closed-loop framework around it. The framework has three pieces. (1) A Unified Instruction Object (UIO): a structured schema of 23 delivery dimensions split into a content axis and a relational axis, resolved from an upstream user-state estimate via a rule-based, confidence-gated pipeline. (2) A dual time-scale adaptation loop: a fast loop that detects per-turn quality degradation and adjusts the next turn's delivery, and a slow loop that learns per-user Dirichlet priors over delivery dimensions across sessions and blends them with rule-based defaults. (3) The Experience Quality Score (EQS), a shared reward combining instruction compliance, behavioral engagement, and (when available) subjective feedback, used to drive both loops. However, the evaluation is simulation-based: 1,094 single- and multi-turn conversations built by pairing DEAP-grounded affective states with PRISM-derived prompts, across Claude and GPT models, comparing the full system (C0) against a conditioned-only ablation (C2) and an unconditioned baseline (C4). The reported findings are that UIO conditioning makes delivery systematically controllable across affective/cognitive states (RQ1), that the full system improves measured quality outcomes over both baselines with large effect sizes (RQ2), and that the two loops operate on their intended timescales (RQ3).

Key strengths. The reframing of delivery as a distinct personalization axis is well-motivated and clearly differentiated from existing static "personality/personalization" settings. The engineering is careful: pre-registered sensitivity checks, Holm correction, two model families with cross-model agreement reported, bootstrap CIs on effect sizes, full hyperparameters, prompts, and even compute cost. The limitations section is candid, particularly about the use of simulated rather than recruited users.

Key weaknesses. The central quality-improvement claims rest on an evaluation loop that is largely self-referential: the reward (EQS), the adaptation signal (SFE), and the simulated user's satisfaction ratings are all derived from overlapping LLM-judged or feature-based sources, and the SFE shares the base model with the generator. There are no human participants, so "interaction quality" is ultimately a property of the simulator's assumptions. As a result, the strongest claims (large quality gains) are not as well-supported as the controllability and mechanism claims.

**Audience:**

Yes

**Audience Explanation:**

Delivery/style personalization for LLMs is a timely and under-explored angle, and the paper offers several things the community can build on regardless of the evaluation concerns: a concrete, reusable parameterization of delivery, a dual time-scale adaptation design that cleanly separates within-session correction from cross-session preference learning, and a fully specified simulation harness (DEAP × PRISM pairing, calibration procedure, prompts, hyperparameters, cost).

**Broader Impact Concerns:**

I dont think a separate Broader Impact Statement is strictly required.

**Claims And Evidence:**

No

**Claims Explanation:**

The controllability claim (RQ1) is convincing. The MANOVA effects, the per-feature decomposition, and the cross-model agreement (r = 0.80) make a clear case that UIO conditioning produces systematic, intended variation in delivery features, and that this is not merely a length effect. Similarly, the mechanism claims (RQ3): weakest-dimension targeting at 96–98%, monotonic slow-loop divergence are internally consistent and adequately evidenced. These are genuine, if somewhat engineering-level, results.

However, the headline claim, stated in the abstract as systematically outperforming the baseline "across all measured quality outcomes" with effect sizes d = 0.61–1.26, is where the evidence does not yet match the strength of the claim. The difficulty is circularity in the evaluation:
- The reward that the system optimizes (EQS) includes a compliance term produced by the SFE, an LLM critic. The fast loop's adaptation direction also comes from the SFE. So the system is trained/adapted toward, and then measured against, signals from the same evaluator family.
- The "interaction quality" outcomes in RQ2 (EQS, SFE overall fit, and six SFE dimensions) are themselves SFE/EQS quantities. Showing that a system optimized for EQS/SFE achieves higher EQS/SFE is close to tautological; the large effect sizes are partly a measure of how directly the system targets its own objective.
- The simulated user's subjective ratings are computed from compliance, supportiveness, clarity, etc. and features that overlap with what the SFE rewards so the "subjective feedback" arm does not provide an independent check either.
- The SFE "uses the same base model as the generation call". A model scoring its own outputs tends to inflate agreement and shares failure modes, which further weakens the outcome measures as evidence of genuine quality.

Beyond this circularity, there is a more fundamental gap: even setting aside whether the evaluation is self-referential, the reward contains no notion of correctness at all. EQS is built entirely from delivery signals (compliance with the style instructions, engagement, subjective comfort), so the system is optimized to make the user feel good about a response, never to check whether the response is right. A reply can top every EQS dimension: warm, well-paced, emotionally attuned, while the advice it carries is wrong, and the framework would score that as a success and reinforce it. The risk is compounded because this is a multi-turn setting, where task accuracy is already known to degrade as conversations lengthen [1].

**Requested Changes:**

1. The central quality-improvement claim is currently supported by measures (EQS, SFE) that the system is optimized toward, scored by an evaluator that shares the generator's base model, with simulated "subjective" ratings built from overlapping features. Maybe re-run the RQ2 outcome evaluation with an evaluator that is independent of the generator (a different model family, a trained classifier, or — best — human raters), and report agreement between the independent evaluator and the SFE.

2. A small human-subjects validation, even ~15 participants on a subset of conditions, would substantially raise confidence that the simulated quality gains track real perceived quality. Given the paper's framing around user experience, this is the most valuable possible addition.

3. The EQS reward has no correctness component, and the multi-turn setting is one where task accuracy is known to be fragile [1]. The reward is built entirely from delivery signals (compliance with the style instructions, engagement, and subjective comfort) and contains no measure of whether the response is actually correct. By construction, the system is rewarded for telling the user what feels good rather than what is true: a response can score at the top of every EQS dimension: warm, well-paced, emotionally attuned, while giving advice that is wrong, and the framework would treat that as a success and reinforce it. This concern is heightened by the fact that lost-in-multi-turn-conversation degradation is itself an active problem, and existing solutions to it (e.g., [2, 3]) could plausibly be embedded into this framework to anchor delivery adaptation to a correctness signal. At a minimum, some discussion of these references and of how the framework guards against correctness degradation across turns should be included.

Refs.

[1] LLMs get lost in multi-turn conversation, ICLR2026

[2] Style Amnesia: Investigating Speaking Style Degradation and Mitigation in Multi-Turn Spoken Language Models

[3] Multi-Turn Reasoning When Context Arrives in Pieces: Scalable Sharding and Memory-Augmented RL

---

### Review · Reviewer_zFTL · 2026-07-13

**Summary Of Contributions:**

The paper treats response delivery, including structure, tone, and relational stance, as a personalization target distinct from content. It proposes:
1. A 23-dimensional UIO that maps an externally estimated user state to delivery constraints.
2. A fast loop that detects quality drops and adjusts selected delivery dimensions.
3. A slow loop that learns cross-session preferences using Dirichlet priors.
4. The EQS, combining compliance, engagement, and subjective feedback.
5. A simulated evaluation using DEAP affect labels, PRISM prompts, and GPT and Claude models.

The framing is clear, the system is thoroughly documented, and the distinction between short-term state and long-term preferences is well motivated. The ablation analysis also shows that static conditioning accounts for 74 to 91% of the reported improvement.

The main weakness is circular evaluation. EQS is both the optimization target and a headline outcome. The SFE drives the fast loop and evaluates its outputs, while the simulated ratings reward features directly controlled by the UIO. The unconditioned baseline also receives no user-state information. Consequently, the experiments show that the system satisfies its own delivery targets, but provide limited evidence of improved user experience. The adaptation study is also based on only five simulated subjects.

**Audience:**

Yes

**Audience Explanation:**

LLM personalization, affect-conditioned generation, and inference-time adaptation are relevant to parts of the TMLR audience. The framework may also be useful as a reference design for future work on adaptive conversational systems.

**Broader Impact Concerns:**

The manuscript would benefit from a dedicated broader impact discussion because the framework infers and stores sensitive affective and relational information. Users should be informed when such conditioning is active and should be able to inspect, modify, and delete their profiles.

**Claims And Evidence:**

No

**Claims Explanation:**

I think the experiments only partially support the claims.
RQ1 convincingly shows that UIO conditioning changes measurable delivery properties. However, this mainly demonstrates instruction following.

RQ2 does not convincingly support the broad claim that the method improves interaction quality. Conditioned systems are evaluated against instructions they receive, while the baseline does not receive those instructions. The subjective outcomes are also generated by a simulator built around the same features the system manipulates. The evidence therefore supports the narrower claim that the system better satisfies its own delivery targets within its simulator.

RQ3 shows that the implementation behaves as designed, but not necessarily that it learns useful preferences. The 96 to 98% targeting result is largely built into the SFE prompt, which explicitly requests adjustments to the weakest dimensions. Fast-loop recovery lacks a no-adjustment control and may reflect regression to the mean. Slow-loop divergence shows that priors change, but not that they move toward correct user preferences.

**Requested Changes:**

- Add a state-in-prompt baseline that receives the same UserState information in natural language without the proposed framework.
- Evaluate quality independently of the target UIO, preferably through blinded human comparisons.
- Compare fast-loop adjustments against matched no-adjustment and random-adjustment controls.
- Clarify how EQS and its compliance component are computed for the unconditioned baseline.
- Evaluate or qualify robustness across thinking and non-thinking models, since the UIO, evaluator calibration, and adaptation thresholds may not transfer unchanged across model generations or reasoning modes.
- Expand the conclusion to summarize the main findings, distinguish demonstrated controllability from simulation-based quality claims, and state the appropriate scope of the contribution.